# Neural Discovery in Mathematics:
# Do Machines Dream of Colored Planes?

Konrad Mundinger [1 2]   Max Zimmer [1 2]   Aldo Kiem [1 2]   Christoph Spiegel [1 2]   Sebastian Pokutta [1 2]

## Abstract

We demonstrate how neural networks can drive mathematical discovery through a case study of the Hadwiger-Nelson problem, a long-standing open problem at the intersection of discrete geometry and extremal combinatorics that is concerned with coloring the plane while avoiding monochromatic unit-distance pairs. Using neural networks as approximators, we reformulate this mixed discrete-continuous geometric coloring problem with hard constraints as an optimization task with a probabilistic, differentiable loss function. This enables gradient-based exploration of admissible configurations that most significantly led to the discovery of two novel six-colorings, providing the first improvement in thirty years to the off-diagonal variant of the original problem (Mundinger et al., 2024a). Here, we establish the underlying machine learning approach used to obtain these results and demonstrate its broader applicability through additional numerical insights.

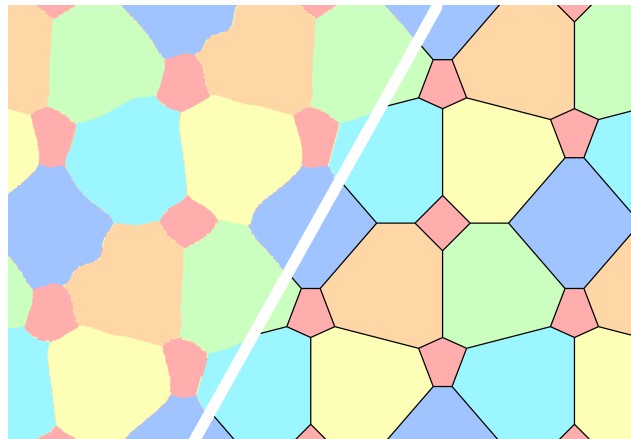

Figure 1: Neural network output (left) that inspired our formal construction (right) of a novel six-coloring of the plane. Red points avoid all distances between 0.418 and 0.657 while other colors avoid unit distances. The neural network was trained through gradient descent on a continuous and differentiable relaxation of these geometric constraints.

## 1. Introduction

The discovery of mathematical constructions is fundamental to theoretical mathematics, where proving the existence of objects with certain properties can advance our understanding of abstract structures. Traditional computational methods have aided the search for such constructions through exhaustive searches, optimization, and heuristics, while recent advances in Machine Learning (ML) offer new ways to explore mathematical structures and guide intuition. Purely discrete problems can be approached through discrete and combinatorial optimization and purely continuous problems through numerical methods, yet many important theoretical problems involve both discrete and continuous aspects. These mixed discrete-continuous problems have remained particularly challenging for traditional approaches.

The Hadwiger-Nelson (HN) problem perfectly illustrates these challenges: determine the minimum number of colors needed to color the Euclidean plane so that no two points at unit distance share the same color. This number is often referred to as the *chromatic number of the plane* $\chi(\mathbb{R}^2)$. Despite its simple statement, this problem combines discrete choices (color assignments), continuous geometry (the Euclidean plane), and hard constraints (unit-distance requirements) in a way that has resisted resolution for over seventy years. While SAT solvers and discrete optimization techniques have contributed to finding lower bounds, discovering new colorings has remained challenging.

We present a framework that reformulates this geometric coloring problem as a continuous optimization task by introducing probabilistic colorings and a differentiable loss function. This continuous formulation enables us to use

---

[1]Department for AI in Science, Society, and Technology, Zuse Institute Berlin, Germany [2]Institute of Mathematics, Technische Universität Berlin, Germany. Correspondence to: Konrad Mundinger <mundinger@zib.de>, Christoph Spiegel <spiegel@zib.de>.

*Proceedings of the 42nd International Conference on Machine Learning*, Vancouver, Canada. PMLR 267, 2025. Copyright 2025 by the author(s).

neural networks (NNs) as universal function approximators to explore the solution space computationally. The networks return numerical approximations of colorings that are equally capable of representing regular patterns or unexpected, irregular solutions (largely) without any strong built-in assumptions about symmetry or periodicity.

Our method serves as a tool for mathematical discovery, operating across a spectrum spanning from fully automated approaches to human-assisted pattern recognition and formalization. Importantly, numerical results do not constitute formal proofs but rather provide guidance that must be formalized. In practice, our approach often discovers clearly interpretable patterns and tessellations that generate insights and candidate constructions, which can be turned into formal results through careful mathematical analysis or, in select cases with sufficient freedom in the solution space, through more automated computational approaches.

**Contributions.** The main contribution of this work is a description of a continuous optimization framework for geometric coloring problems that uses NNs and probabilistic colorings, enabling gradient-based exploration of solution spaces without strong prior assumptions or discretization of the underlying space. This framework was essential in the discovery of several novel colorings that imply improvements for variants of the HN problem:

1. We obtained two 6-colorings of the plane for a variant of the original problem where five colors must avoid unit distances while the sixth color avoids a different specified distance. These colorings significantly expand the known range of realizable distances to $[0.354, 0.657]$ from the previous best of $[0.415, 0.447]$ due to Hoffman & Soifer (1996) and Soifer (1994b).

2. We obtained a 5-coloring covering all but 3.74% of the plane where each of the five colors must avoid unit distances, improving on the previous best known bound of around 4.01% due to Parts (2020b). We also obtained a 14-coloring covering all but 3.46% of three-dimensional space.

3. We extended the range of monochromatic triangles avoidable when coloring the plane with a given number of colors, another well-known variant. The previous best bounds are due to Aichholzer & Perz (2019).

Formal descriptions of the colorings underlying the first result were published separately by Mundinger et al. (2024a), while papers describing the remaining results are currently in preparation. In the present article, besides describing the underlying framework that enabled this mathematical discovery, we provide additional numerical insights and include negative results for other variants where the framework indicated no avenue for improvement.

**Related Literature.** Recent years have seen growing interest in applying ML to mathematical discovery, from automated theorem proving to finding novel constructions. We focus on the latter, specifically on approaches that use ML to discover mathematical objects with desired properties. For a broader perspective, we refer to recent surveys by Wang et al. (2023a); Krenn et al. (2022); Williamson (2023).

Several approaches have emerged for using ML in combinatorial discovery. Most closely related to our work, Karalias & Loukas (2020) developed an unsupervised GNN-based framework that uses probabilistic relaxations and differentiable loss functions to solve graph optimization problems. While they focus on purely discrete problems like maximum clique finding, we consider a problem with an inherent continuous structure. Fawzi et al. (2022) solved a discrete optimization problem using a reinforcement learning approach, from which they were able to derive a faster matrix multiplication algorithm. Similarly, Wagner (2021) studied the ability of a reinforcement-learning approach to construct discrete counterexamples to conjectures in extremal combinatorics. Building on this framework, Roucairol & Cazenave (2022) studied conjectures in spectral graph theory. However, Parczyk et al. (2023) found that this approach was largely outperformed by well-established local searches while studying the Ramsey multiplicity problem. This finding was at least in part supported by the work of Mehrabian et al. (2024). Charton et al. (2024) combine both approaches and propose `PatternBoost`, which trains an NN on the output of a local search and which was also used by Bérczi & Wagner (2024). Taking a markedly different approach, Romera-Paredes et al. (2024) propose `FunSearch`, which uses LLMs to guide greedy search through evolutionary optimization of generated code. Novikov et al. (2025) very recently improved upon this approach through `AlphaEvolve`. Other notable approaches of applying ML to mathematical problems in general include Alfarano et al. (2024) and Trinh et al. (2024).

Our approach differs from the ones used in these results, since we use ML primarily as a tool for mathematical discovery requiring further interpretation and formalization, similar to work of Davies et al. (2021) on knot invariants, which laid the foundation for Davies et al. (2024) to prove a formal statement with significant additional mathematical effort. This approach of using Artificial Intelligence (AI) to guide mathematical intuition, while still requiring rigorous proofs, represents a challenging but underserved direction in AI-assisted mathematics - one that demands expertise in both ML and pure mathematics.

The paradigm of approximating functions by NNs which take coordinates as input is known as *implicit neural representations* (Park et al., 2019; Mescheder et al., 2019; Chen & Zhang, 2019). Our approach adopts this strategy while intro-

ducing a loss function that penalizes deviations from desired geometric properties. This connects our method to physics-informed neural networks (Raissi et al., 2019; Cuomo et al., 2022; Wang et al., 2023b), where NNs approximate solutions to differential equations through unsupervised residual minimization. Berzins et al. (2024) recently extended these ideas to geometric constraints.

Finally, our approach fits within the broader *AI4Science* paradigm, where ML is revolutionizing scientific discovery in domains such as protein structure prediction (Jumper et al., 2021) or the earth sciences (Pauls et al., 2024).

## 2. The Hadwiger-Nelson Problem

First posed in 1950, the HN problem has become one of the most famous open questions in combinatorial geometry and graph theory, with a rich history of partial results and increasingly sophisticated approaches. For a comprehensive treatment of the problem, see Soifer (2009).

Lower bounds for $\chi(\mathbb{R}^2)$ are typically established by constructing unit distance graphs – graphs whose vertices can be embedded as points in the plane in such a way that edges only connect vertices whose embeddings are a unit distance apart. The Moser spindle establishes that at least four colors are needed (Moser & Moser, 1961) and this bound remained the best known until the breakthrough proof of de Grey (2018) established that $\chi(\mathbb{R}^2) \geq 5$. His construction, originally involving a unit distance graph with 20,425 vertices, sparked intense and largely computational efforts, including through a Polymath project, to simplify and reduce this graph (Heule, 2018; Exoo & Ismailescu, 2020; Parts, 2020a; Polymath, 2021; Gwyn & Stavrianos, 2020).

Upper bounds on the other hand are commonly established through explicit colorings of the plane. There exists a large number of distinct seven-colorings that avoid monochromatic pairs at unit distance, the first of which used a tiling of congruent regular hexagons and was already discovered in 1950 according to Soifer (2009). This upper bound of $\chi(\mathbb{R}^2) \leq 7$ has remained unchanged since. While computational methods have proven highly successful in establishing lower bounds, there have been surprisingly few systematic computational approaches to constructing new colorings. The only one we are aware of was the use of SAT solvers to color coarsely discretized versions of specific metric spaces as part of the discussions during the Polymath project (Polymath, 2021), highlighting the need for techniques that can explore the continuous solution space more directly.

Given the challenge of closing the gap between the lower and upper bounds of the original problem, several natural variants have been proposed. Solutions to these variants can provide additional insight into the underlying structure and complexity of the original problem.

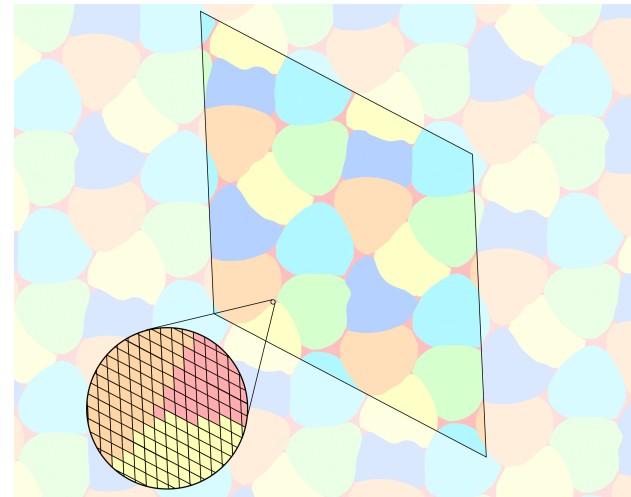

Figure 2: The formalized almost 5-coloring obtained using the discretization procedure described in Algorithm 1. It is periodic and constant on small parallelograms as indicated by the magnified section. The red color represents the part that covers 3.74% of $\mathbb{R}^2$ and that needs to be removed.

**Variant 1: Almost coloring the plane.** A natural question concerns how much of the plane needs to be removed before the remainder can be colored with six colors without monochromatic unit-distance pairs. Pritikin (1998), with a later refinement due to Parts (2020b), established that 99.985% of the plane can be colored with six colors while maintaining this property. This result also effectively bounds the density of a potential counterexample: any unit distance graph requiring seven colors would need to have order at least 6,993, see Lemma 1 by Pritikin (1998) for a proof. The construction uses intersecting pentagonal rods and our NN approach, when applied to the original HN problem with six colors, consistently recovered structures remarkably similar to these pentagonal constructions. Parts (2020b) also studied the same question for fewer than six colors and we likewise applied our approach in this setting with a modification in the form of a Lagrangian term. This resulted in an improvement for the 5-color variant, where our construction, see Figure 2, successfully covers all but 3.7356% of the plane. This improves on the previous best value of 4.0060% due to Parts (2020b), who explicitly asked in his paper for new approaches to push the value below the mark of 4%. See Section 4.1 for further details.

**Variant 2: Avoiding different distances.** Another natural generalization of the problem considers colorings where different colors avoid different distances. More precisely, we say that an $k$-coloring of the plane has *coloring type* $(d_1, \ldots, d_k)$ if color $i$ does not realize distance $d_i$. This defines a natural 'off-diagonal' variant of the original problem, where finding a coloring of type

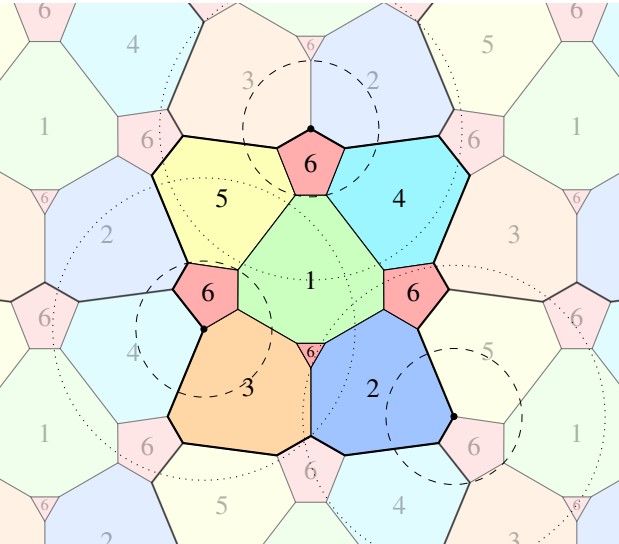

Figure 3: Formalized version of a neural network-generated 6-coloring of the plane where no red points appear at distance 0.45 and no other color has monochromatic unit-distance pairs. The dotted circles have unit distance radius while the dashed circle have radius 0.45.

$(1,1,1,1,1,1)$ would establish $\chi(\mathbb{R}^2) \leq 6$. Stechkin found a coloring of type $(1,1,1,1,1/2,1/2)$ (Raiskii, 1970b) and Woodall (1973) constructed one of type $(1,1,1,1/\sqrt{3},1/\sqrt{3},1/\sqrt{12})$. Erdős asked for the *polychromatic number of the plane* $\chi_p(\mathbb{R}^2)$ (Soifer, 1992a), that is the smallest number of colors $k$ such that there exist some $d_1,\ldots,d_k$ and a coloring of type $(d_1,\ldots,d_k)$. So far no better bounds than $4 \leq \chi_p(\mathbb{R}^2) \leq 6$ are known. Soifer (1992b) found the first six-coloring with a non-unit distance in only one color, having type $(1,1,1,1,1,1/\sqrt{5})$. Hoffman & Soifer (1993) later found colorings of type $(1,1,1,1,1,\sqrt{2}-1)$. These constructions are part of a family that realizes $(1,1,1,1,1,d)$ for any $\sqrt{2}-1 \leq d \leq 1/\sqrt{5}$ (Hoffman & Soifer, 1996; Soifer, 1994b). This led Soifer (1994a) to pose the 'still open and extremely difficult' (Nash & Rassias, 2016) problem of determining the continuum of six colorings – that is the set of all $d$ for which a six-coloring of type $(1,1,1,1,1,d)$ exists.

Using our approach we discovered two novel colorings of the plane that extend the continuum of six colorings, providing the first improvement in thirty years. The first coloring is parameterized by $d$ and valid for type $(1,1,1,1,1,d)$ as long as $0.354 \leq d \leq 0.553$, see Figure 3. The second coloring is constant and covers the range of $0.418 \leq d \leq 0.657$, see Figure 1 and Figure 11. Together they significantly expand the range of distances for which colorings are known to exist. A complete description of these constructions is given by Mundinger et al. (2024a). We also explored the

polychromatic number computationally but found no way to improve the upper bound, see also Section 4.2.

**Variant 3: Coloring $n$-dimensional space.** This variant extends the HN problem to higher dimensions, seeking to determine the chromatic number $\chi(\mathbb{R}^n)$ of $n$-dimensional space while maintaining the constraint that points at unit distance must receive different colors. A general lower bound of $\chi(\mathbb{R}^n) \geq n+2$ was established by Raiskii (1970a). The three-dimensional case has received particular attention: Nechushtan (2002) proved $\chi(\mathbb{R}^3) \geq 6$ using a graph on approximately 400 vertices, which de Grey (2020) later refined to a 59-vertex construction. Regarding upper bounds, Coulson (2002) famously showed $\chi(\mathbb{R}^3) \leq 15$, a bound that Soifer (2009) conjectures to be tight. Our approach successfully found colorings using 15 colors as well as, applying the techniques from the first variant, a 14-coloring covering all but 3.4622% of $\mathbb{R}^3$, see also Section 4.3.

**Variant 4: Avoiding triangles.** Finally, one can ask how many colors are needed to avoid monochromatic triples of points each at unit distances to each other. The answer in this case is actually quite simple: coloring the plane with alternating parallel and half-open stripes of width $\sqrt{3}/2$ using two colors, taking care to include the correct parts of the border of each strip in the color, avoids any monochromatic such triangle. One may again consider a natural off-diagonal version of this problem in which the triples form triangles with prescribed side lengths $1$, $a$, and $b$ for some $0 \leq a, b \leq 1$ satisfying $a + b > 1$. Note that the (degenerate) case $a = 0$ and $b = 1$ corresponds to the original HN problem. A conjecture of Erdős, Graham, Montgomery, Rothschild, Spencer, and Straus (Graham, 2003) states that three colors should always be sufficient, though so far little progress has been made towards that conjecture with some cases, where $a + b$ is close to 1, still requiring seven colors, see Aichholzer & Perz (2019). Currier et al. (2024) recently also proved that any two-coloring contains three-term arithmetic progressions, which can be viewed as a degenerate triangle with $a = b = 0.5$. Using our approach, we were able to extend the previous best bounds in several regions, see Figure 4 and Section 4.4.

Many other variants of the HN problem exist, including generalizations to other metric spaces, colorings avoiding arbitrary unit-distance graphs, or colorings avoiding several distinct differences in each color. Our framework should be largely applicable to these variants with minimal modifications, providing a systematic tool for exploring potential constructions.

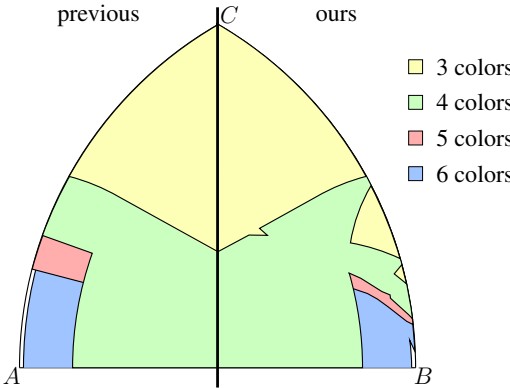

**Figure 4:** Location of the vertex $C$ of triangles $ABC$ with side $AB$ of length 1 for which a coloring avoiding monochromatic copies of such a triangle is known to exist for between three and six colors. The previously known results due to Aichholzer & Perz (2019) are shown on the left and the results including our improvements on the right. Note that Figure 7 in (Aichholzer & Perz, 2019) contained an error that significantly under-reported the size of the four-color region and that has been corrected here.

## 3. Methodology

In this section we develop the formulation of the HN problem as a continuous optimization problem using probabilistic colorings. The original problem asks whether there exists a function $g \colon \mathbb{R}^2 \to \{1, \ldots, c\}$ such that

$$g(x) \neq g(y) \quad \forall x, y \in \mathbb{R}^2 \text{ with } \|x - y\|_2 = 1. \quad (1)$$

We relax the constraints by replacing the discrete color assignment with a probabilistic one. Instead of assigning a fixed color to each point, we assign a probability distribution over the $c$ colors, represented by a mapping $p \colon \mathbb{R}^2 \to \Delta_c$, where $\Delta_c$ is the $c$-dimensional probability simplex. The inner product $p(x)^T p(y)$ then naturally captures the probability that points $x$ and $y$ receive the same color when independently sampling from their respective distributions.

Given such a probabilistic coloring $p$, we define a loss function measuring the violation of the unit distance constraint in $[-R, R]^2$ through

$$\mathcal{L}_R(p) = \int_{[-R,R]^2} \int_{\partial B_1(x)} p(x)^T p(y) \, d\nu(y) \, d\mu(x), \quad (2)$$

where $\partial B_r(x) = \{y \in \mathbb{R}^2 \mid \|x - y\| = r\}$ denotes the euclidean sphere of radius $r$ around $x$ and $\nu = \mathcal{U}(\partial B_1(x))$ as well as $\mu = \mathcal{U}([-R, R]^2)$ are uniform distributions over $\partial B_1(x)$ and $[-R, R]^2$, respectively. The inner integral represents the probability of a conflict between a fixed point $x$ and a randomly sampled point $y$ at unit distance from it, while the outer integral averages this over all points in

$[-R, R]^2$. While we are ultimately interested in this loss as $R$ tends to infinity, in practice we study it for some fixed (sufficiently large) value of $R$ and aim to solve

$$\arg\min_p \mathcal{L}_R(p). \quad (3)$$

A coloring of a finite box $[-R, R]^2$ does not necessarily extend to a valid coloring of the entire plane $\mathbb{R}^2$. However, our objective is to find patterns within this box that can be extended across the plane.[1]

Clearly, any valid discrete coloring $g$ of the plane can be converted to a probabilistic coloring $p$ through one-hot-encoding, yielding $\mathcal{L}_R(p) = 0$ for all $R > 0$. In the other direction, we have the following simple result.

**Proposition 3.1.** *If $R > 0$ and $p$ is a probabilistic coloring with $\mathcal{L}_R(p) = 0$, then the coloring given by $g(x) = \arg\max\big(p(x)\big)$ satisfies Equation* (1) *for almost all pairs* $\{(x, y) \in [-R, R]^2 \times \mathbb{R}^2 \mid \|x - y\| = 1\}$.

*Proof.* Let $X \subseteq [-R, R]^2$ denote the set of points $x$ for which the inner integral $\int_{\partial B_1(x)} p(x)^T p(y) dy$ is non-zero and for any given $x \in [-R, R]^2$ let $Y(x) \subseteq \partial B_1(x)$ denote the set of points $y$ for which the support of $p(y)$ is not disjoint from $p(x)$, i.e., the points $y$ with $p(x)^T p(y) > 0$. By basic properties of the Lebesgue measure and since $p(x)^T p(y)$ in the definition of $\mathcal{L}_R(g)$ is non-negative as a dot product of probability vectors, $X$ has measure zero in $[-R, R]^2$ and so does $Y(x)$ in $\partial B_1(x)$ for any $x \in [-R, R]^2 \setminus X$. It follows that $\{(x, y) \mid x \in X, y \in \partial B_1(x)\} \cup \{(x, y) \mid x \in [-R, R]^2 \setminus X, y \in Y(x)\}$, likewise has measure zero with respect to the uniform distribution over $\{(x, y) \in [-R, R]^2 \times \mathbb{R}^2 \mid \|x - y\| = 1\}$. $\square$

The minimization of Equation (2) over probabilistic colorings therefore provides a path to finding colorings for the original HN problem through Proposition 3.1, assuming a suitably parameterized family of functions that ideally can approximate arbitrary colorings.

### 3.1. Parameterizing the space of probabilistic colorings

We approximate probabilistic colorings $p$ through NNs $p_\theta$ of fixed architecture and size with trainable parameters $\theta$, turning Equation (3) into the finite-dimensional optimization problem $\arg\min_\theta \mathcal{L}_R(p_\theta)$. By virtue of the universal approximation property (Cybenko, 1989) and their inherent spectral bias towards low-frequency functions (Rahaman et al., 2019), NNs provide both the expressiveness needed

---

[1]Instead of sampling $x$ uniformly from a finite box, one could sample from a Gaussian distribution with variance corresponding roughly to the box size $R$. In limited tests, this alternative did not yield significantly different or improved constructions.

to capture complex spatial patterns and a natural tendency towards structured, interpretable solutions.

We employ a standard multilayer perceptron which encodes geometric structure by operating on coordinates, making this a geometric learning approach despite its simple architecture. Following best practices for implicit neural representations (Sitzmann et al., 2020), we implement $p_\theta$ with sine activation functions. The networks we used consisted of two to four hidden linear layers with 32 to 256 neurons each, directly mapping coordinates to color distributions through a softmax activation after the final layer.

### 3.2. Batched gradient descent

To minimize the loss function, we employ gradient descent on the NN parameters $\theta$. The main challenge lies in approximating $\nabla_\theta \mathcal{L}_R(p_\theta)$, which involves nested integrals that we approximate through Monte Carlo sampling. Specifically, we sample $n$ center points $x_i \sim \mathcal{U}\left([-R, R]^2\right)$ and, for each center $x_i$, sample $m$ points $y_{ij} \sim \mathcal{U}\left(\partial B_1(x_i)\right)$, obtaining the estimate

$$\nabla_\theta \mathcal{L}_R(p_\theta) \approx \nabla_\theta \left[ \frac{1}{nm} \sum_{i=1}^{n} \sum_{j=1}^{m} p_\theta(x_i)^T p_\theta(y_{ij}) \right]. \quad (4)$$

### 3.3. Considerations for each of the four variants

Let us outline how the formulation in Equation (2) can be adapted to the four variants described in Section 2, including necessary adjustments to the NN and sampling procedures already described.

**Variant 1: Almost coloring the plane.** To formalize the idea of an *almost coloring*, we introduce an additional color, in which no conflicts are penalized. We extend the codomain of $p_\theta$ to $\Delta_{c+1}$, with the last entry corresponding to the additional color. Formally, we then want to solve the constrained optimization problem

$$\underset{p_\theta}{\arg\min} \, \mathcal{L}_R(p_\theta^c)$$

$$\text{s.t.} \int_{[-R,R]^2} p_\theta(x)_{c+1} \, d\mu(x) \leq \delta$$

for some pre-determined $\delta > 0$, where $p_\theta^c$ refers to the first $c$ components of $p_\theta$. More precisely, we want to determine the minimum $\delta$ for which the answer to this problem is zero. In practice, we solve the Lagrangian relaxation

$$\mathcal{L}_R^\lambda(p_\theta) = \mathcal{L}_R(p_\theta^c) + \lambda \int_{[-R,R]^2} p_\theta(x)_{c+1} \, d\mu(x). \quad (5)$$

While in theory there exists a corresponding $\lambda$ for each $\delta$, we simply treat $\lambda$ as a hyperparameter controlling the trade-off between using the additional color and maintaining valid colorings.

Almost colorings contain some inherent freedom in the the solution space, as we can simply increase the size of the additional color by a small amount if needed, so we designed a fully automated pipeline to obtain rigorously verified colorings. After identifying periodic structures in an initial trained NN in the form of a tesselation using a parallelogram, we retrain a separate NN to enforce precisely that periodicity. We then discretize the output and eliminate any unit-distance conflicts through an iterative discrete heuristic. The periodic extension of this discrete solution yields a continuous almost coloring that provably satisfies all unit-distance constraints, see also Algorithm 1 and Figure 2.

---

**Algorithm 1** Automated almost-coloring formalization

---

1. **Initial training.** Train $p_\theta : \mathbb{R}^2 \to \Delta_{c+1}$ to minimize Equation (5) on a large enough box $[-R, R]^2$.

2. **Periodicity extraction.** Determine two vectors $v_1, v_2 \in \mathbb{R}^2$ with $0 \ll \angle(v_1, v_2) \ll \pi$ such that the coloring (largely) consists of tiling the parallelogram

$$\mathcal{P} = \{\alpha v_1 + \beta v_2 : \alpha, \beta \in [0, 1)\}$$

along the lattice $\Lambda = \{ n_1 v_1 + n_2 v_2 : n_1, n_2 \in \mathbb{Z} \}$.

3. **Periodicity-constrained retraining.** Form the invertible change-of-basis matrix $M = [v_1 \; v_2] \in \mathbb{R}^{2 \times 2}$. Prepend the mapping $x \mapsto M^{-1}x \pmod 1$ to $p_\theta$, which enforces exact periodicity over $\Lambda$, and retrain.

4. **Discrete almost-coloring.** Discretize $\mathcal{P}$ into $kl$ copies of $\{\alpha v_1/k + \beta v_2/l : \alpha, \beta \in [0, 1)\}$ and determine a color for each parallelogram pixel by sampling $p_\theta$ at its respective center.

5. **Iteratively fix remaining conflicts.** Determine a discrete mask in which conflicts need to be avoided around each parallelogram pixel to obtain a formal coloring. Iteratively reduce any remaining conflicts by solving an auxiliary minimum edge cover problem and recoloring some parallelograms. After a fixed number of rounds, resolve any remaining conflicts by recoloring with the additional color $c + 1$.

---

**Variant 2: Avoiding different distances.** For colorings of type $(d_1, \ldots, d_c)$ we modify the loss formulation to

$$\sum_{k=1}^{c} \int_{[-R,R]^2} \int_{\partial B_{d_k}(x)} p_\theta(x)_k \, p_\theta(y)_k \, d\nu_k(y) \, d\mu(x), \quad (6)$$

with $\mu = \mathcal{U}\left([-R,R]^2\right)$ and $\nu_k = \mathcal{U}\left(B_{d_k}(x)\right)$, i.e., we now need to choose the points $y$ from the sphere $\partial B_{d_k}(x)$ of radius $d_k$ around $x$ for each color $k$. Minimizing the loss function finds probabilistic colorings that attempt to avoid distance $d_k$ for color $k$. Note that we recover Equation (2) for $d_1 = \ldots = d_c = 1$.

This model can also be extended to handle sets or ranges of distances for some or all colors. To achieve this, we redefine the domain of the candidate functions $g$ to include the current distance. Given distance ranges $[d_k^{\min}, d_k^{\max}]$ for each color $k$ for example, we set

$$p : \mathbb{R}^2 \times \underset{k=1}{\overset{c}{\times}} [d_k^{\min}, d_k^{\max}] \to \Delta_c. \qquad (7)$$

By integrating Equation (6) over this expanded domain, we average the loss across all distances within the specified range for each color by sampling points on $c$ circles, corresponding do the distances $d_1, \ldots, d_c$. Note that this increases the effective batch size from $n\,m$ to $n\,m\,c$. This approach allows exploration of a larger part of the solution space instead of just a single type.

**Variant 3: Coloring $n$-dimensional space.** To extend to higher dimensions, it suffices to integrate over $\mathbb{R}^n$ and letting $\partial B_1(x)$ represent the sphere in the appropriate space. We are particularly interested in the three-dimensional case. Apart from adapting the input dimension of $g_\theta$ as well as sampling in the higher dimensional space, this case is analogous to the two dimensional cases. The question of almost colorings arises naturally in higher dimensions as well. Here, the automated verification procedure described in Algorithm 1 is particularly useful, as visualization and interpretation of colorings gets much more challenging in higher dimensions.

**Variant 4: Avoiding triangles.** In order to avoid monochromatic triangles with side lengths $1$, $a$, and $b$, we replace the integrand $p_\theta(x)_k\,p_\theta(y)_k$ in Equation (2) with

$$\sum_{k=1}^{c} p_\theta(x)_k\,p_\theta(y)_k\,\big(p_\theta(z_1)_k + p_\theta(z_2)_k\big)/2 \qquad (8)$$

where $z_1$ and $z_2$ are the two candidates for the the point $z$ such that $x$, $y$, and $z$ form a triangle with $\|x - y\| = 1$, $\|x - z\| = a$ and $\|y - z\| = b$, assuming $a + b > 1$.

In this variant we sample center points $x \in [-R, R]^2$ and proximity points $y$ on the unit circle as before. We then obtain the two candidates $z_1$ and $z_2$ for the third point of the triangle by calculating the intersection points of the circles of radius $a$ and $b$ around $x$ and $y$, respectively. As in Variant 2, the domain of the function $p$ can be extended to $\mathbb{R}^2 \times [a_{\min}, a_{\max}] \times [b_{\min}, b_{\max}]$ to allow for continuous ranges of $a$ and $b$ within one network and we integrate the loss over the intervals $[a_{\min}, a_{\max}]$ and $[b_{\min}, b_{\max}]$.

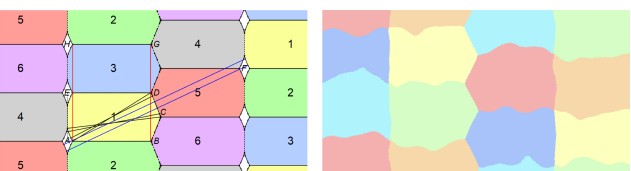

Figure 5: The almost coloring due to Pritikin (1998) and Parts (2020b) (left) and the coloring suggest by our approach (right). The wavy lines are caused by the inherent degrees of freedom present in many colorings.

## 4. Experimental results

In this section we present our numerical results for the original HN problem and its four variants. We implemented our approach in PyTorch, updating the parameters using the Adam optimizer (Kingma & Ba, 2015) with a learning rate schedule linearly decaying (Li et al., 2020) from an initial value between $10^{-3}$ and $10^{-4}$. We typically used between 16,384 and 65,536 training steps, sampling 1,024 to 4,096 points for $x$ and 1 to 32 points for $y$ on the circle(s) around $x$ during each step, resulting in $10^6$ to $10^{10}$ total samples during training. A single such run takes between two and 20 minutes on an NVIDIA V100 GPU, though adequate results can be achieved in under 10 minutes on a standard laptop CPU. Due to the heuristic nature of our approach and the need to tune hyperparameters, we did however perform thousands of runs for each variant, particularly when exploring parameter spaces in the off-diagonal and triangle variants.

Our initial experiments focused on six-color solutions in the classical setting. The method consistently recovered patterns resembling Pritikin's construction, providing early validation of our approach, see Figure 5. However, these results also revealed that NNs often introduce irregularities such as wavy lines, reflecting inherent freedoms in the solution space. This becomes even more pronounced for seven colors, see Figure 9 in the appendix. These findings highlight that our method can successfully discover valid colorings, but formalizing the discovered patterns into rigorous mathematical constructions may still require substantial additional effort. Additionally, the method's inability to suggest valid colorings with fewer than seven colors provides additional numerical evidence that the chromatic number of the plane may in fact be seven. The code needed to reproduce our results and the constructions obtained using Algorithm 1 are available at https://github.com/ZIB-IOL/neural-discovery-icml25.

### 4.1. Variant 1: Almost coloring the plane

Applying the Lagrangian-modified loss function described in Section 3.3 to six colors, we achieved conflict rates

Table 1: We compare our results with the best known for almost coloring the plane, reporting the fraction of points that must be removed to color the remainder without unit-distance conflicts. *Prior* results are due to Croft (1967) and Parts (2020b). *Numerical* values were obtained by evaluating a trained model on a parallelogram with periodical boundary conditions. The *formalized* values were obtained by applying Algorithm 1.

| # colors | 1 | 2 | 3 | 4 | 5 | 6 |
|---|---|---|---|---|---|---|
| **prior** | 77.04% | 54.13% | 31.20% | 8.25% | 4.01% | 0.02% |
| **numerics** | 77.07% | 54.21% | 31.34% | 8.29% | **3.60%** | 0.03% |
| **formalized** | 77.13% | 54.29% | 31.51% | 8.52% | **3.74%** | 0.04% |

of around 0.03%, slightly higher than those reported by Parts (2020b). The optimization process proved sensitive to the Lagrangian term $\lambda$, but consistently recovered known constructions once a good range (typically between $10^{-3}$ and $10^{-1}$) was found. The discovered patterns for one to four colors closely match known constructions due to Croft (1967), see Figure 10 in the appendix. The five-color case exhibits patterns similar to those in the construction described by Parts (2020b), but the numerical value of around 3.60% suggested the possibility of an improvement. The formalized construction does in fact attain a significantly improved value of 3.7356%. Complete numerical results for $k = 1, \ldots, 6$ colors along with the formalized values obtained using Algorithm 1 are reported in Table 1.

### 4.2. Variant 2: Avoiding different distances

We report the numerical results on colorings of type $(d_1, \ldots, d_6)$ with a particular focus on the cases $d_i = 1$ for either $i \in \{1, \ldots, 5\}$ or $i \in \{1, \ldots, 4\}$.

**Colorings of type** $(1, 1, 1, 1, 1, d)$. Following Section 3, we included the free distance $d$ in the domain of the coloring function, exploring the interval $[0.2, 2.2]$. This approach not only allows optimization over a continuous range of distances but also reveals smooth transitions between colorings, see Figure 12 in the appendix, which proved crucial in identifying a broader family of valid solutions. Figure 6 shows the conflict rates as a function of $d$ for sixteen individual runs and their pointwise minimum. While previous work established valid colorings for $d \in [\sqrt{2} - 1, 1/\sqrt{5}]$ (orange dashed lines), our constructions extend this range to $[0.354, 0.657]$ (blue dashed lines). Additional regions of interest appear at $d \approx 1$, corresponding to the original HN problem and $d \approx 1.6$, where conflict rates fall below 1%.

**Colorings of type** $(1, 1, 1, 1, d_1, d_2)$. Analogously, we treated the last two colors as free distances and augmented the domain of the NNs accordingly. The resulting colorings remain continuous in both $d_1$ and $d_2$. As shown in Figure 7,

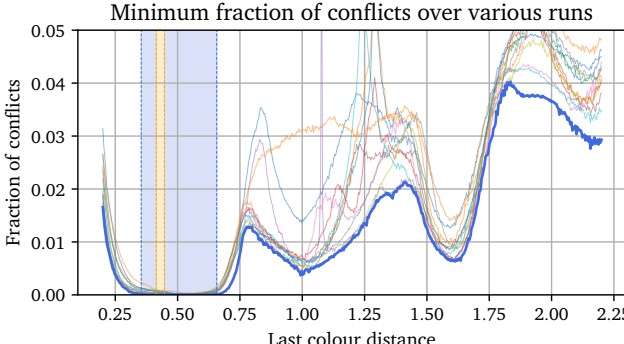

Figure 6: Fraction of conflicting points versus the free distance $d$ in colorings of type $(1, 1, 1, 1, 1, d)$. The thick blue line shows the minimum over multiple runs (thin lines). Orange and blue shaded regions indicate the previously known and our extended ranges of valid colorings, respectively.

we found three regions of minimal conflicts: two symmetric regions near $(d_1, 1)$ and $(1, d_2)$ with $d_1, d_2 \approx 0.5$, recovering our earlier $(1, 1, 1, 1, 1, d)$ colorings, and an additional region near $(0.5, 0.5)$ corresponding to a construction similar to Stechkin's, see Figure 12.

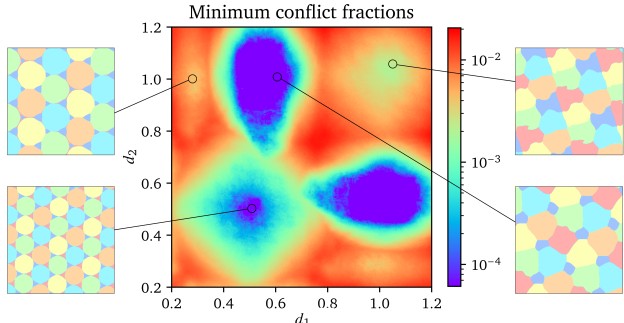

Figure 7: The fraction of conflicts in $(1, 1, 1, 1, d_1, d_2)$ colorings as a function of $d_1$ and $d_2$ (center, showing minimum over sixteen networks). Three regions of minimal conflicts emerge: symmetric regions near $(d_1, 1)$ and $(1, d_2)$ with $d_1, d_2 \approx 0.5$, and one near $(0.5, 0.5)$. Sample colorings from these regions are shown in the surrounding plots.

**Polychromatic Number.** We investigated potential five-color solutions of type $(d_1, \ldots, d_5)$ by including the distances in the domain of the NN and determining the best candidates for such a type after training using first- and second-order methods. Our result achieved a minimal conflict rate of 4.9% with distances $d_1 = 1$, $d_2 \approx d_3 \approx 1$, and $d_4 \approx d_5 \approx 0.56$, see Figure 13 in the appendix. Despite extensive optimization, our inability to find a conflict-free five-color solution provides additional evidence that the polychromatic number may indeed be six. More details are provided in Appendix C.

### 4.3. Variant 3: Coloring $n$-dimensional space

Our method successfully found (near) conflict-free 15-color solutions, consistent with the bound established by Coulson (2002). Our attempt to improve this bound directly to 14 was not successful. We also studied the almost coloring variant in three dimensions, which resulted in numerical solutions with a conflict rate of around 2.3%. Applying a three-dimensional modification of Algorithm 1 yielded a formal coloring covering all but 3.4622% of $\mathbb{R}^3$, though further refinements improving this value seem possible. More details are provided in Appendix D.

### 4.4. Variant 4: Avoiding triangles

We applied our approach to study triangles with sides of length 1, $a$, and $b$ satisfying $a + b > 1$. For four and five colors, our NNs consistently discovered constructions based on stripes and hexagonal tessellations, similar to those used by Aichholzer & Perz (2019). However, the numerical results suggested that alternative scalings of these basic patterns could yield improvements beyond those previously known, see the suggested patterns in Figure 8. We were able to formalize some of those based on stripes, obtaining several new bounds that expand the regions where three to five colors suffice, as already shown previously in Figure 4. This significantly reduces the area of the parameter space still requiring seven colors, though further improvements seem possible.

Beyond further resolving the parameter space with respect to the number of required colors both numerically (Figure 8) as well as via formalized patterns (Figure 4), the numerical results also provide evidence against the conjecture of Graham (2003) that three colors should always suffice: our experiments consistently required more colors as one side of the triangle became shorter while the other became closer to unit length, suggesting these cases (which more closely resemble the original HN problem) may be fundamentally more challenging.

## 5. Discussion

Our framework provides a novel approach to exploring mathematical structures through continuous optimization, leading to several concrete improvements to long-standing open problems. Beyond just being applicable to variants of the HN problem, we believe that it naturally extends to other problems in discrete geometry and extremal combinatorics. For instance, graph-theoretic problems involving discrete structures can often be reformulated through graph limits, where the objective becomes optimizing over continuous maps $[0,1]^2 \to [0,1]$. Our approach of using NNs as universal function approximators combined with suitable loss functions could be directly applied in this setting. The

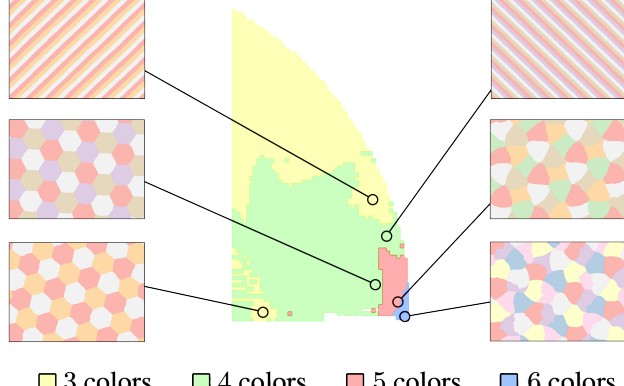

□ 3 colors  □ 4 colors  □ 5 colors  □ 6 colors

Figure 8: Numerical results showing the achievable color bounds when avoiding monochromatic triangles, together with a selection of found colorings. We trained thousands of networks on different sub-areas of the above region and say that a point can be achieved with three, four, five, or six colors if the top 3% of runs for that point reported that less than 0.1% of sampled triangles were monochromatic in the argmax coloring derived from the trained NNs.

framework could also potentially be extended to handle non-differentiable losses using auxiliary 'critic' or 'value' networks that learn to approximate these losses, similar to techniques used in adversarial and reinforcement learning.

We also experimented with other function families such as Fourier bases and Voronoi diagrams; however, NNs consistently performed best, due to their expressiveness, ease of training, and ability to represent complex functions with relatively few parameters. The approach required a substantial amount of tuning, particularly in the choice of architecture and hyperparameters. Beyond the usually relevant parameters such as the learning rate, the Lagrangian penalty term was particularly important: although scale-sensitive, it led to stable and consistent results once a good range was found. The size of the bounding box was less critical, as long as it was large enough to avoid trivial colorings.

We believe that the combination of AI-aided constructions with human insight is highly beneficial, leveraging the strengths of automated exploration alongside human interpretation and mathematical intuition. We also emphasize that, in the case of the almost colorings, we developed a fully automated pipeline that no longer requires human intervention. This automation was already helpful in two dimensions, but became essential for higher-dimensional variants of the problem, where manual interpretation becomes increasingly challenging. More broadly, this work demonstrates how continuous relaxations and gradient-based optimization can bridge the gap between ML and mathematical discovery, potentially opening new paths to approaching long-standing open problems in mathematics.

## Acknowledgements

Research reported in this paper was partially supported through the AI4Forest project, which is funded by the German Federal Ministry of Education and Research (BMBF; grant number 01IS23025A) and the Deutsche Forschungsgemeinschaft (DFG) through the DFG Cluster of Excellence MATH+ (grant number EXC-2046/1, project ID 390685689) as well as Sonderforschungsbereich (SFB) Transregio 154 (grant number 239904186). C.S. would also like thank Marijn Heule for his FoCM 2024 talk on HN that inspired this research project and Sagdeev Arsenii for suggesting the triangle variant of this problem. The authors would also like to thank Tibor Szabó for suggesting sampling from a Gaussian distribution instead of from a finite box as well as the referees for many valuable suggestions.

## Impact Statement

This work demonstrates how ML can drive scientific and mathematical discovery, using NNs to explore solutions to the HN problem and related combinatorial geometry challenges. Our approach bridges discrete and continuous mathematical domains, enabling researchers to explore problem spaces more broadly and deeply than traditional methods allow. By reformulating complex mathematical problems as optimization tasks, we provide tools for intuition-driven discovery that retain interpretability while accelerating progress. While the immediate focus is on theoretical mathematics, the broader applicability of ML methods similar to our approach has the potential to transform diverse fields that require navigating hybrid discrete-continuous solution spaces, such as materials science, network optimization, and beyond. The societal impact of this paradigm shift could significantly enhance our ability to address long-standing scientific and engineering problems.

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

# A. Coloring the plane with seven colors

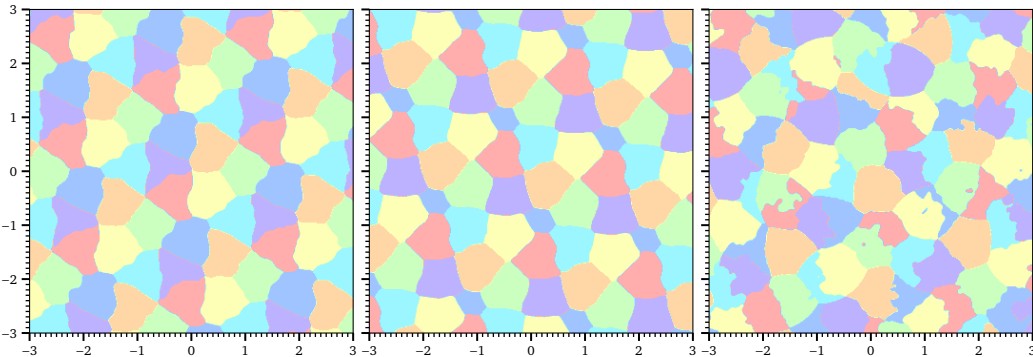

Figure 9: Colorings found for the Hadwiger-Nelson problem with seven colors. Left and middle: Rather structured colorings. Right: Less structured coloring.

## B. Details on variant 1: Almost coloring the plane

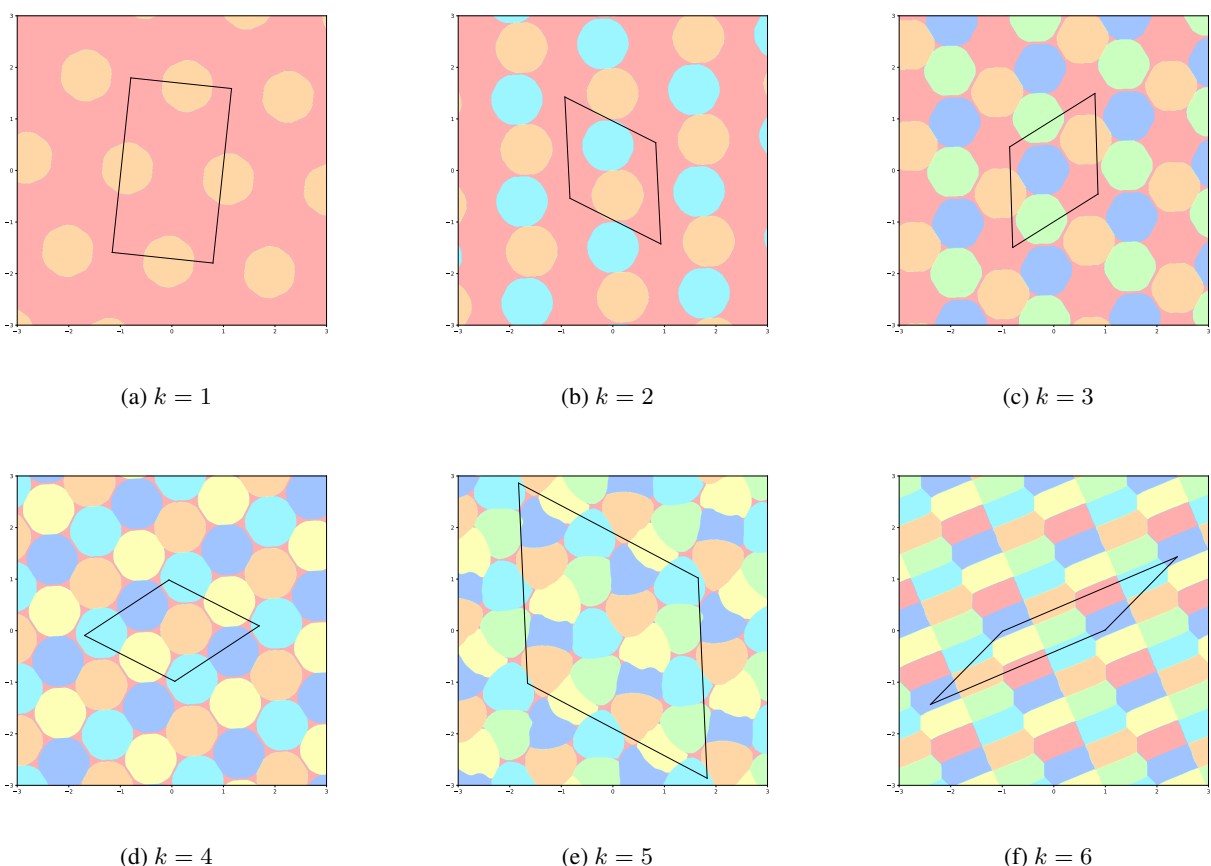

(a) $k = 1$                (b) $k = 2$                (c) $k = 3$

(d) $k = 4$                (e) $k = 5$                (f) $k = 6$

Figure 10: Almost colorings for $k = 1, \ldots, 6$ obtained through Algorithm 1. The parallelograms indicate the fundamental domain along which periodicity is enforced.

In the case of almost colorings, we designed a fully automated pipeline to generate rigorously verified colorings which tessellate the plane using parallelograms and are piecewise constant on smaller parallelograms, as outlined in Algorithm 1. To infer the directions of periodicity, we evaluate the trained models from step 1 along radial lines from the origin and translate these lines by increasing offsets. A direction is categorized as a direction of periodicity if the similarity between the original and translated evaluations exceeds a threshold after a distance $d$. From all such directions, we then extract two vectors that have minimal norm and are sufficiently separated in angle. These vectors span the fundamental parallelogram which tiles the plane.

We then retrain a new model enforcing periodicity on this parallelogram and subdivide it into smaller sub-parallelograms (which we refer to as 'cells') and treat the coloring as constant within each one. For each cell $c_{base}$, we determine all the cells which contain points that are at unit distance to any point in $c_{base}$ by computing the cells that intersect with unit circles at the corners of $c_{base}$. We then turn the unit distance constraint into the stricter (but discrete) constraint of not having cells with the same color if they contain unit distance pairs.

Finally, in step 5, we employ a heuristic to completely eliminate unit distance conflicts while minimizing the use of the "bonus" color. While one could trivially assign the "bonus" color to all conflicting cells, we instead construct an auxiliary conflict graph whose nodes are conflicting cells and whose edges connect same colored cells at approximate unit distance. By computing a minimum edge cover and setting the selected cells to the "bonus" color, we effectively resolve all remaining conflicts.

## C. Details on variant 2: Avoiding different distances

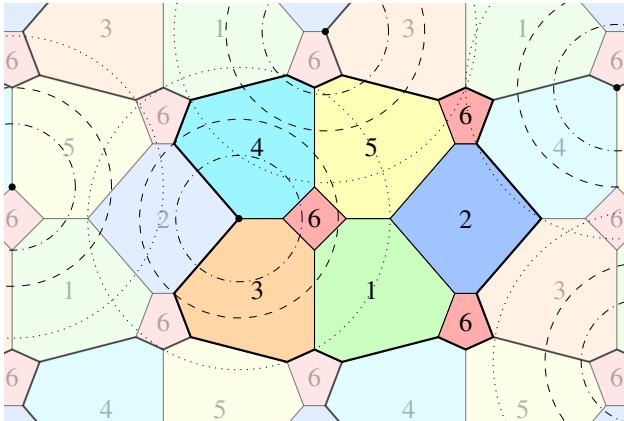

Figure 11: A 6-coloring of the plane where no red points appear at any distance between $0.418$ and $0.657$ while no other color has monochromatic unit-distance pairs. The dotted circles have unit distance radius, while the dashed circles have radius $0.657$ and the dash-dotted circles have radius $0.418$. Shown is the formalized version, which was inspired by the NN outputs, see also Figure 1.

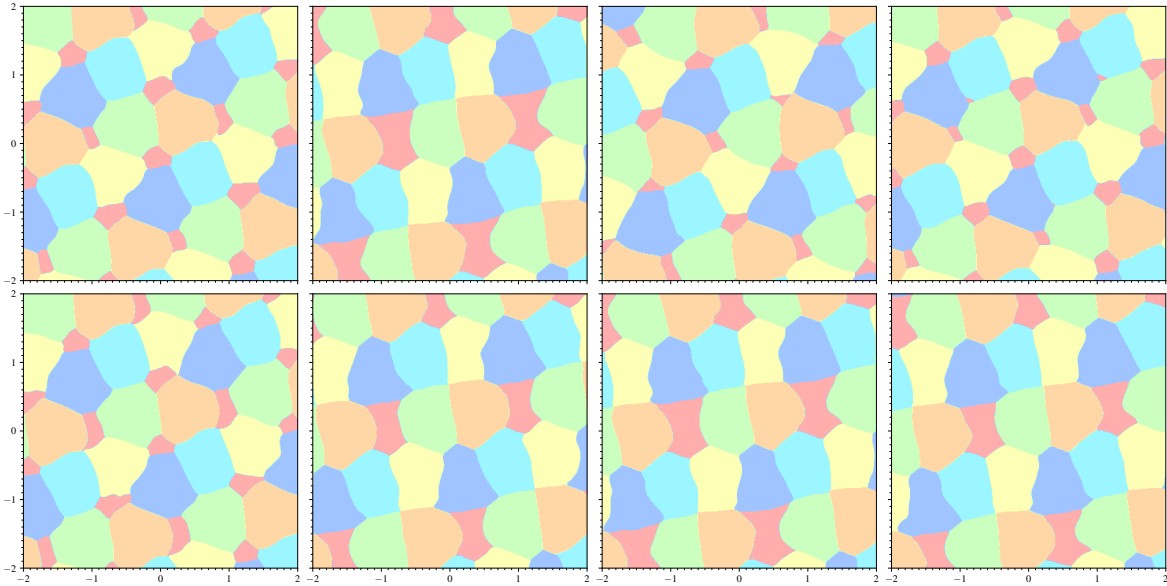

Figure 12: A more detailed view of the colorings generated by a single model for different values of the last distance. Upper row (from left to right): $d = 0.4$, $d = 0.6$, $d = 0.8$, $d = 1.0$. Lower row (from left to right): $d = 1.2$, $d = 1.4$, $d = 1.6$, $d = 1.8$.

The polychromatic number of the plane is the minimum number of colors needed such that no color realizes all distances in its monochromatic pairs. While known to be at most 6, finding a five-color solution would constitute a significant improvement. Our search focused on colorings of type $(d_1, \ldots, d_5)$, where we set $d_1 = 1$ without loss of generality. Our approach sampled distances from $[0.2, 2.2]$ and included them directly in the domain of the NN. Rather than performing an expensive grid search over the four remaining distances, we employed first- and second-order optimization methods to find optimal values after training. This optimization led to our best result with distances $d_1 = 1$, $d_2 \approx d_3 \approx 1$ and $d_4 \approx d_5 \approx 0.56$, achieving a minimal conflict rate of approximately 5% (see Figure 13). This suggests that the polychromatic number of the plane may indeed be six.

Note that when including the distances in the domain of the function of the coloring, we conceptually parametrize a whole

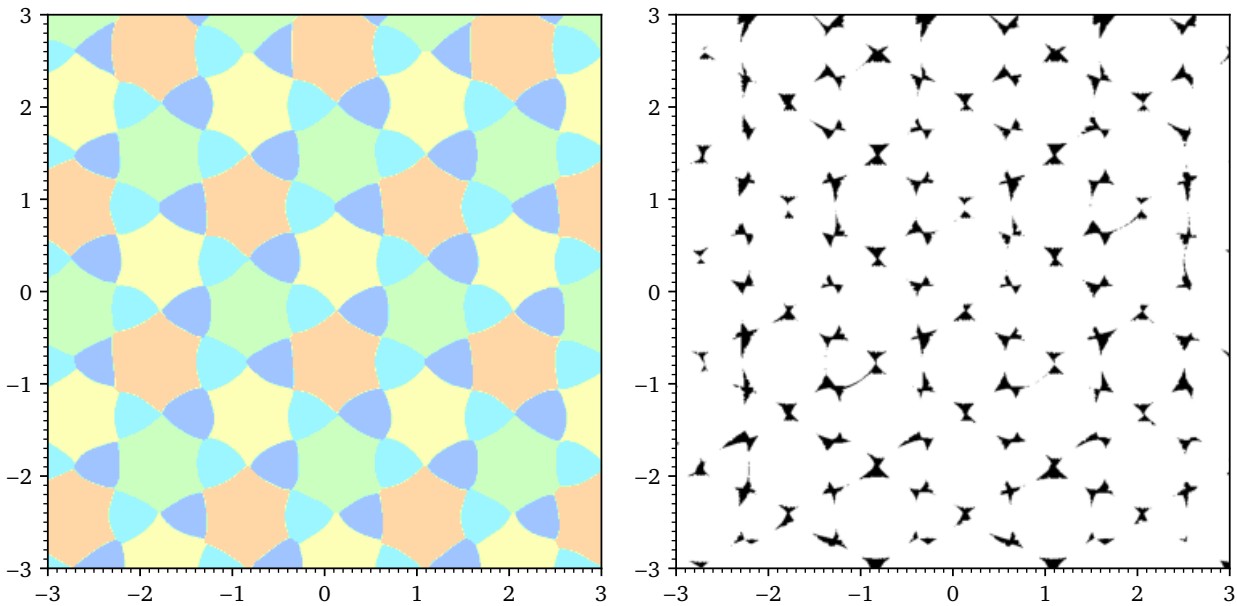

Figure 13: The best five-coloring with $d_1 = 1$, $d_2 \approx d_3 \approx 1$ and $d_4 \approx d_5 \approx 0.56$ and approximately 5% of conflicts in a box of size $[-3, 3]^2$.

family of implicit neural representations. Instead of viewing this as a function of both coordinates and distances as in Equation (7), we can view it as a function-valued function (i.e., an operator):

$$G : \bigtimes_{k=1}^{c} [d_k^{\min}, d_k^{\max}] \to \left\{ g \mid g : \mathbb{R}^2 \to \Delta_c \right\}. \tag{9}$$

This perspective connects to the rapidly growing field of operator learning (Kovachki et al., 2024). DeepONets (Lu et al., 2021) are particularly suitable architectures for such mappings, and have proven effective in the numerical treatment of differential equations (Wang et al., 2021; Mundinger et al., 2024b).

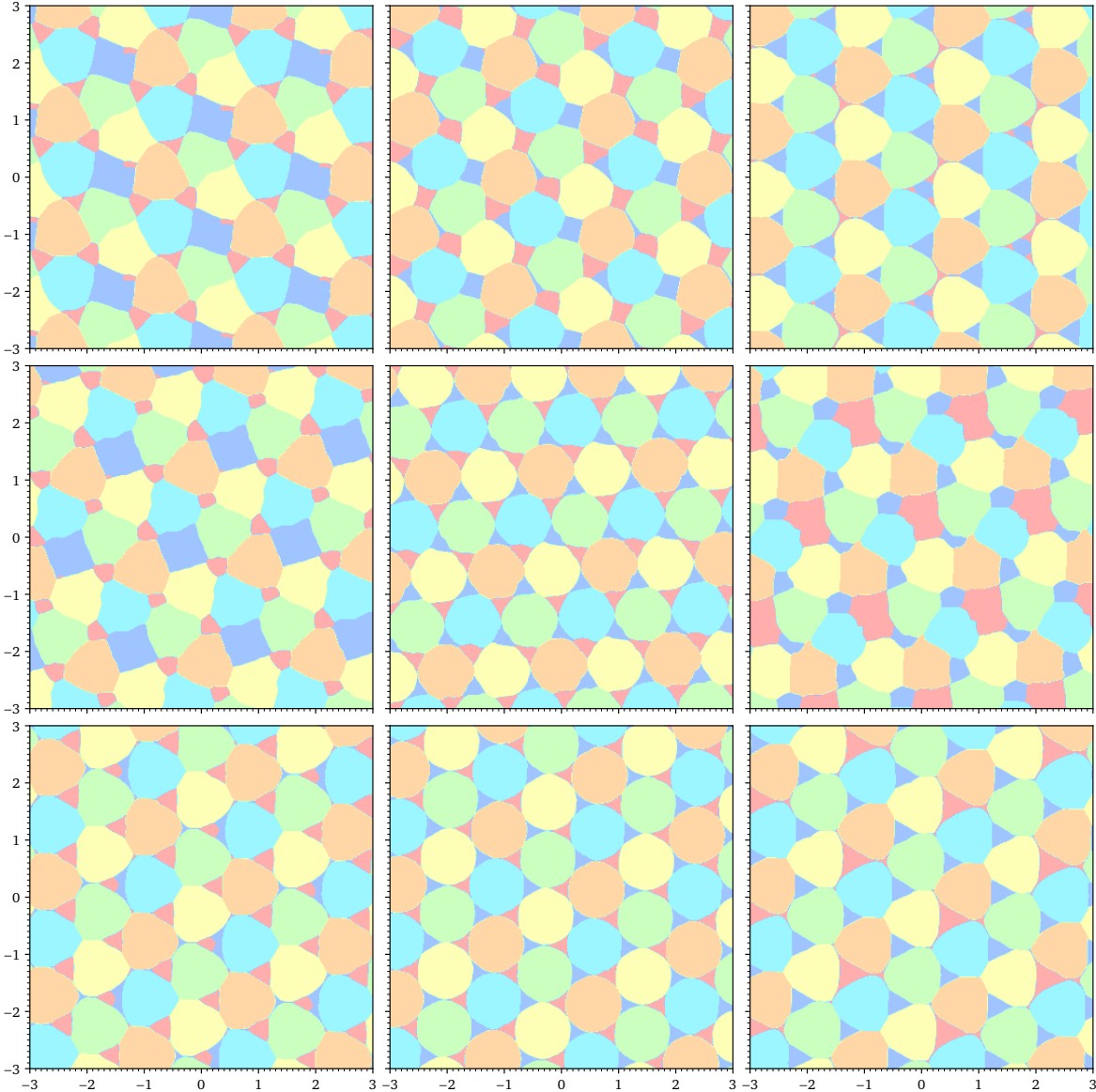

Figure 14: Three different distance configurations generated by the same model: $(d_1, d_2) = (1, 0.4)$ (left), $(0.5, 0.5)$ (middle), and $(0.6, 1.0)$ (right). The bottom two rows show patterns similar to Stechkin's construction at $(0.5, 0.5)$, while the outer columns reveal patterns that led to our formalized constructions.

# D. Details on Variant 3: Coloring three-dimensional space

Visualizing and thus also interpreting and formalizing a three-dimensional coloring is much more challenging than in the two dimensional case. To better understand the structure of colorings, we extracted polyhedral regions from the NN output and explored them in interactive plots. One of these colorings is shown in Figure 15.

Since our numerical experiments did not suggest the existence of a 14-coloring without unit distance conflicts, we applied an adapted version of Algorithm 1 to construct an almost-14-coloring of space in a fully automated fashion. It covers all but 3.46% of $\mathbb{R}^3$, though it seems that this value might be improved through a finer discretization.

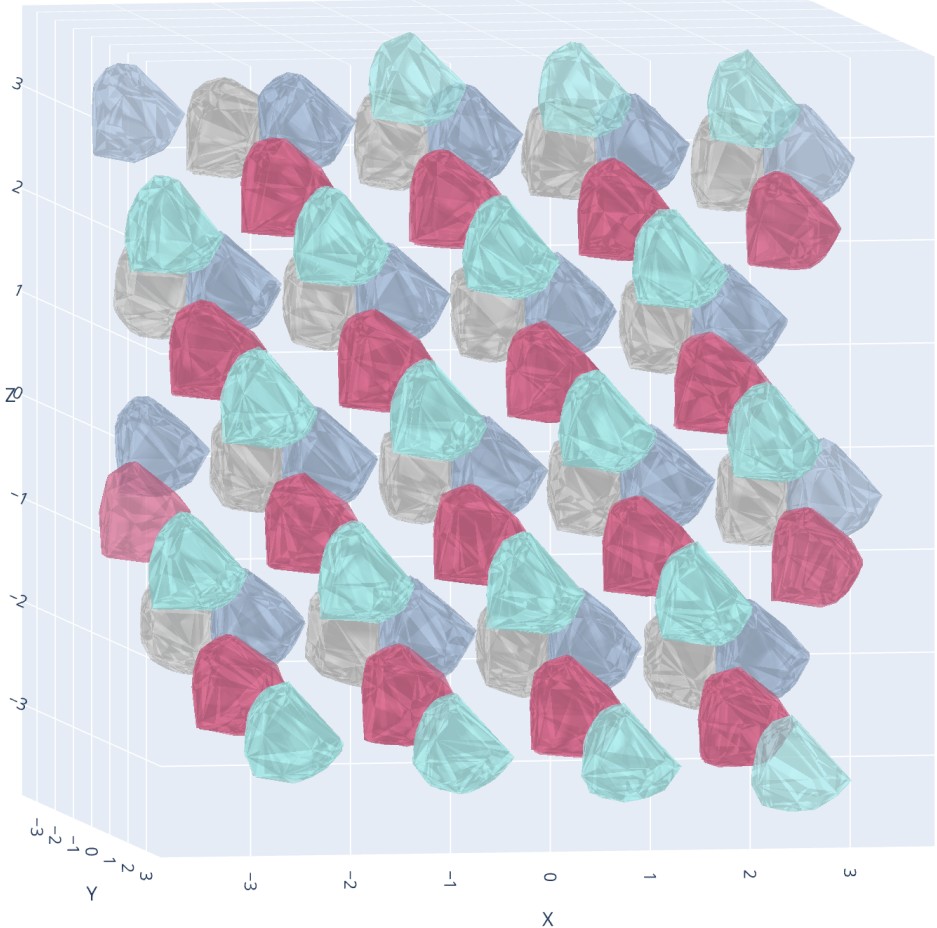

Figure 15: A three-dimensional coloring found by our approach. We only show four out of the fourteen colors.

# E. Details on Variant 4: Avoiding triangles

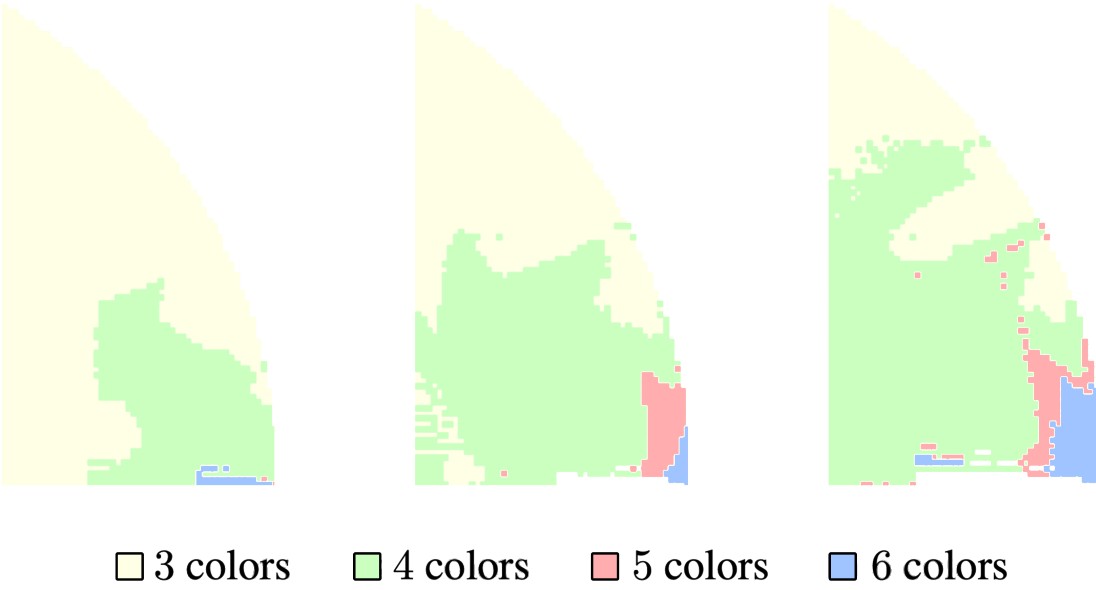

□ 3 colors    □ 4 colors    □ 5 colors    □ 6 colors

Figure 16: Numerical results showing the achievable color bounds when avoiding monochromatic triangles. We trained thousands of networks on different sub-areas of the above region and say that a point can be achieved with three, four, five, or six colors if the top 3% of runs for that point reported that less than 1% (left), 0.1% (middle), and 0.01% (right) of sampled triangles were monochromatic in the argmax coloring derived from the trained NNs.

# F. Different p-norms

Our approach extends to different norms in $\mathbb{R}^n$. The loss function remains the same, while the respective ball $B_1(x)$ is replaced by the $L_p$-ball $B_p(x) = \{y \in \mathbb{R}^n \mid \|x - y\|_p \leq 1\}$. To generate samples from the $L_p$-ball, we use the probabilistic method proposed by Barthe et al. (2005). The cases $p = 1$ and $p = \infty$ in $\mathbb{R}^2$ are relatively straightforward to derive, as there exist colorings using four colors by arranging the balls in a regular grid. However, in particular the case $p = 1$ produces visually interesting results, as not only the trivial constructions are found but variations thereof, see Figure 17.

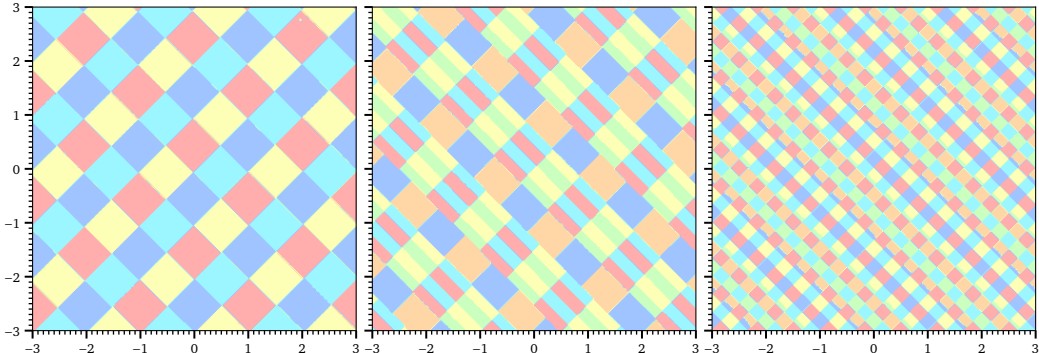

Figure 17: Colorings found for the $L^1$ norm. Left: The trivial grid coloring. Middle and right: Colorings which utilize the degrees of freedom present in the coloring.

