# OpenReview forum: "Neural Discovery in Mathematics: Do Machines Dream of Colored Planes?"
_ICML.cc/2025/Conference — ICML 2025 oral_

### Official Review · Reviewer_dBz6 · 2025-03-04

**Overall Recommendation:** 4

**Summary:**

This work demonstrates how neural networks can help mathematical discovery, focusing on the Hadwiger-Nelson problem, which seeks the minimum number of colours needed to colour the Euclidean plane while avoiding unit-distance monochromatic pairs. By reformulating this mixed discrete-continuous problem as an optimization task with a differentiable loss function, the authors leverage neural networks to explore admissible colourings. This approach led to the discovery of two novel six-colorings.

## Update after rebuttal:

This paper presents an interesting contribution to the field of AI for Mathematics. It is generally well written, with only a few minor typos, which the authors have acknowledged and stated they have corrected.

There are, however, areas that could be improved. For instance, some statements come across as quite strong—for example, in lines 056–060: "Rather than attempting to obtain proofs that are fully automated end-to-end, our method serves as a tool for mathematical discovery by providing the right intuition about potential constructions." This claim may not always hold and could benefit from a more balanced or cautious phrasing. The authors mentioned that will address this point.

Another noteworthy aspect of the paper is the discussion on using neural network outputs to guide the construction of HN contracts. The authors mention that this process still requires significant manual experimentation (see question/answer 4), which is an important limitation to highlight. The authors mentioned that will add this paper in the paper.

Overall, I find the paper interesting and well-executed. In light of the rebuttal and improvements, I have increased my score from 3 to 4.

**Claims And Evidence:**

The main evidence for this paper is based on a cited work listed as Anonymous, Anonymous Journal, 2024, indicating that the original idea originates from the same authors. The anonymous paper is included in the supplementary material. In lines 069–072, the authors mention that the anonymous work has been formally verified and published in a venue specializing in combinatorial geometric problems. Therefore, I see no reason why the anonymous paper was not consistently referenced, leading the reviewers to fail to realize that the authors are the same. This has resulted in a misunderstanding.

**Essential References Not Discussed:**

The authors have included many related works, covering the most important ones. However, I found the paper "A Finite Graph Approach to the Probabilistic Hadwiger-Nelson Problem" by H. Gwyn et al., which appears relevant and could be added.

**Experimental Designs Or Analyses:**

Yes, I checked the experimental designs. They seem correct, but I believe there is room for improvement. For example, since the authors use a neural network to minimize the continuous loss for the Hadwiger-Nelson problem, there is no plot showing that this loss decreases to zero over time. Including such a plot would be helpful.

**Methods And Evaluation Criteria:**

The approach to this problem makes sense; the authors aim to relax the discrete optimization of the Hadwiger-Nelson problem into a continuous format and minimize its loss using neural networks. It would be helpful for them to provide plots of the loss and the results of optimizing the hyperparameters of the neural network. Additionally, it would be beneficial to see an example where this approach is applied to a toy version of the Hadwiger-Nelson problem with a known solution, ensuring its validity before presenting results for the more complex cases.

**Other Comments Or Suggestions:**

A) There are a few typos in the text:
1) At the beginning, the authors refer to 'Neural Networks' as 'NN', so please ensure consistency throughout the text (see lines 049, 090, 420, 423, 442).
2) The same applies to 'Machine Learning'—please maintain consistency (see lines 040, 096, 432, 442, 453).
3) The word 'invariant' appears twice consecutively (see lines 076, 077).
4) There is a typo in the word 'nuber', which should be 'number' (see line 308).
5) 'Hadwiger-Nelson' should be replaced by HN, (see line 443).
6) In the captions (see Figures 1 and 2), you write "Neural Networks" with a capitalized first letter, whereas in the abstract and main text, you sometimes use "neural network" in lowercase and other times "NN". Additionally, in the appendix figures, abbreviations are used inconsistently—for example, in Figure 9, "HN" is used instead of the full term, and similar inconsistencies appear in Figures 12 and 16. Please ensure consistency throughout. In the abstract and captions, use the full term "neural networks", while in the main text, introduce it as "neural networks (NNs)" the first time and then use "NNs" thereafter. Similarly, maintain a consistent approach for other abbreviations like "HN".
7) You forgot the full stop in some cases after the bold text at the beginning of paragraphs (see lines 365, 373, and 380). Please ensure consistency by adding a full stop after the bold text where needed.

B) Many citations reference ArXiv papers, despite the existence of official publications. Since ArXiv papers are not guaranteed to have undergone a peer review process, which is crucial, please replace ArXiv citations with the formal published versions. For example see line 459, the official paper is 'The signature and cusp geometry of hyperbolic knots Davies, A Juhasz, A Lackenby, M Tomasev, N Geometry and Topology volume 28 issue 2024 2313-2343 (24 Aug 2024)'

C) In some parts of the text, you refer to details in the appendices but do not specify the exact section. Please be more specific, as seen in lines 321, 340, and 370. Follow the same approach as in line 383.

D) Please be sure that you have added any relevant literature. For example the paper 'A finite graph approach to the probabilistic Hadwiger-Nelson problem, by H Gwyn et. al.' seems relevant to me. If it's not please explain me.

E) In some parts of the text, references need to be added. For example, in line 316, when referring to Pritikin’s construction, and in line 352, when mentioning Stechkin’s construction, appropriate citations should be included for clarity and proper attribution.

F) It is important to clearly understand your achievement regarding color planes. It would be helpful if you could add a table comparing previous findings with your own results. Could you include this? The table could be placed either in the introduction or the main text.

**Other Strengths And Weaknesses:**

The paper introduces the idea of transforming the Hadwiger-Nelson (HN) problem from discrete optimization to continuous optimization, which is an interesting and, to the best of my knowledge, novel direction. Their findings are also noteworthy. However, while the discrete-to-continuous transformation is valuable, it is not a new concept in general. Regarding the statement in lines 056-060—"Rather than attempting to obtain proofs that are fully automated end-to-end, our method serves as a tool for mathematical discovery by providing the right intuition about potential constructions."—I believe the underlying intuition does not stem from the use of neural networks but rather from the fact that the problem itself is now continuous. As a result, this claim seems somewhat strong and unrealistic, and it would be better to present it in a more cautious and balanced manner.

**Questions For Authors:**

1) I’m a bit confused about the anonymous paper you added and how your work builds upon it. Could you clarify the key differences between your previous paper and the current one?

2) Could you create a toy example of the Hadwiger-Nelson problem where the exact result is known and compare it with the outcome from the neural network? This would help demonstrate that your model performs well even in a simple case where a 'ground truth' exists, ensuring its validity before presenting results for more complex scenarios.

3) Could you create a table comparing the results of previous works with your own? This would make it clearer what you have achieved in comparison to others.

4) In Figure 1, on the left side, the boundaries between different colors are not straight lines, whereas on the right side, they are. How exactly did you construct this? More generally, what method do you use to transition from the continuous to the discrete problem?

**Relation To Broader Scientific Literature:**

From what I understand, this paper makes two main contributions: a) transforming the discrete optimization Hadwiger-Nelson problem into a continuous one and using neural networks to solve it, b) discovering two novel six-colorings of the plane for the off-diagonal Hadwiger-Nelson problem based on their results.

**Theoretical Claims:**

There is one proposition that validates the transformation from discrete optimization to continuous optimization (see lines 239–243), which seems correct.

---

> ### Author Rebuttal · Authors · 2025-03-28
>
> Thank you for your thoughtful review and constructive feedback. Let us first address your questions and then the remaining comments:
>
>
> **Question 1:**
>
> Regarding the anonymous citation, we would like to clarify that the referenced paper—listed as "Anonymous (2024)" and included as supplementary material—was authored by us and presents specific mathematical constructions derived using the methodology introduced in this ICML submission. It has been accepted in a journal specializing in combinatorial and geometric problems, which we view as strong validation of our approach. Our ICML paper focuses on the underlying methodology that led to these results, in a manner similar to  [Davies et al. (2021b)](https://www.nature.com/articles/s41586-021-04086-x.pdf)  and [Davies et al. (2021a)](https://msp.org/gt/2024/28-5/p06.xhtml), where the methodological framework was presented in Nature and the resulting mathematical findings were published separately in more specialized venues. Due to the constraints of the double-blind review process, we anonymized the citation. We are happy to allow our journal publication to be reviewed by the AC/SAC without disclosing our identities to the reviewers.
>
> **Question 2:**
>
> We consider the results covered in Figures 9, 10, and 11 to serve as validation examples, as they recover known constructions. However, we're happy to include specific additional examples with known solutions if that would strengthen the paper. Could you clarify what types of toy examples you have in mind that would be most helpful?
>
> **Question 3:**
>
> We appreciate this suggestion. While we understand the value of direct comparisons, our work presents unique challenges for tabular representation for two key reasons:
>
> 1. The improvements we demonstrate span multiple dimensions that cannot be easily reduced to single-value comparisons.
> 2. We consider formalized, peer-reviewed constructions as the primary benchmark for validating our methodology, rather than simplified metrics.
>
> Nevertheless, we recognize the importance of clearly presenting our contributions relative to previous work. In our revised submission, we will add a comprehensive comparison that highlights our achievements alongside previous approaches. This will provide a clearer picture of our contributions while maintaining the nuance necessary for proper evaluation.
>
> **Question 4:**
>
> Thank you for this important question. The transition from continuous neural network output (left) to formal construction (right) involves careful human interpretation and mathematical formalization. The non-straight boundaries in the neural network output reflect the inherent flexibility in the solution space. We intentionally straightened these boundaries in the formalized version to simplify the mathematical description without compromising validity. For the off-diagonal colorings in Figure 2, this formalization process required about one week of manual experimentation using basic trigonometry to convert the neural network's patterns into precise mathematical expressions. The complexity of this step varies depending on the specific problem variant.
>
>
> **Additional Points:**
>
> * Regarding your statement about intuition not stemming from neural networks but rather from the continuous formulation: We appreciate this perspective, but would like to clarify that neural networks did provide the specific visual patterns that inspired our formal constructions. While one could certainly use alternative function families as universal approximators, the specific patterns produced by our neural network approach directly led to the insights that enabled our formal constructions. We'll revise our presentation to make this relationship clearer and more balanced.
> * We agree that including plots showing the loss decreasing during optimization would strengthen the paper. We'll add plots showing typical convergence patterns for our successful runs, with losses decreasing to values around 1e-7 for seven colors and 1e-6 for six colors (and sometimes as low as 1e-12) when evaluated on a 512 × 512 discretized grid.
> * Thank you for highlighting "*A Finite Graph Approach to the Probabilistic Hadwiger-Nelson Problem*". We are happy to expand the related literature section to include more references regarding improvements of the lower bounds.
> * We'll fix all the typographical errors and inconsistencies you noted.
> * We'll replace arXiv citations with published versions where available, this was definitely an oversight during our submission.
> * We'll add specific section references when referring to appendix materials and we'll add appropriate citations for Pritikin's and Stechkin's constructions whenever mentioned.
>
>
> Thank you again for your thorough feedback, which will help us improve the clarity and rigor of our paper.

---

> > ### Comment · Reviewer_dBz6 · 2025-04-05
> >
> > Thank you so much for your detailed and clear responses to my points.
> >
> > Q1: Regarding the anonymous paper, I understand its contribution. If the AC needs to verify it, they will reach out to you, so there is no need to worry about that.
> >
> > Q2: I appreciate the clarification.
> >
> > Q3: A comparison table would indeed be helpful, as mentioned by the reviewer cNSg in the 'Methods and Evaluation Criteria' section.
> >
> > Q4: Thank you for explaining everything in detail. In my view, the transition from continuous neural network output (left) to formal construction (right), which you described to me, is quite interesting and important. I believe this aspect could be a valuable addition to your paper.
> >
> > Final Note: Regarding my comment in the 'Other Strengths and Weaknesses' section, I still feel that the statement in lines 056-060— "Rather than attempting to obtain proofs that are fully automated end-to-end, our method serves as a tool for mathematical discovery by providing the right intuition about potential constructions."—seems quite strong and may not always hold true. It might be better to soften this claim for a more balanced presentation.
> >
> > Based on your responses to all the reviewers and your consideration of our comments, I am happy to upgrade the score to 4.
> >
> > Thank you again for addressing my points and clarifying the misconceptions!

---

> > > ### Author Response · Authors · 2025-04-09
> > >
> > > Thank you again for your thorough and valuable review and for increasing your score.
> > > We agree with both your and reviewer cNSg's suggestions regarding a better presentation of our results which we will include in the revision of the paper.
> > > We will rework the formulation regarding your final note towards a more balanced presentation; we will also detail how the formalized constructions were obtained.
> > > Thank you very much again for your constructive comments and suggestions, which have greatly helped to make our paper more clear and rigorous.

---

### Official Review · Reviewer_wpnp · 2025-03-07

**Overall Recommendation:** 4

**Summary:**

This paper tackles a combinatorics problem using neural networks, specifically the Hadwiger-Nelson problem, which involves coloring the plane under distance constraints. The problem is continuously relaxed through probabilistic coloring. Inspired by numerical outputs from neural networks, the authors derive a formal construction that advances the field. Additionally, the paper explores multiple variants with different constraints.

**Claims And Evidence:**

The paper is well-written with beautiful visualizations.

**Essential References Not Discussed:**

N/A

**Experimental Designs Or Analyses:**

Seems to be correct.

**Methods And Evaluation Criteria:**

Intuitive and reasonable.

**Other Comments Or Suggestions:**

N/A

**Other Strengths And Weaknesses:**

I believe this paper is a solid scientific contribution and would be relevant to the ML community working on combinatorial optimization.

However, while this paper may inspire new applications, its technical contribution seems limited. Continuous relaxation for combinatorial optimization is a well-established approach, and relaxing the coloring problem into a probabilistic form is a natural choice. While I appreciate the authors' effort in designing formulations for different variants, it is unclear whether these formulations introduce non-trivial insights that could meaningfully influence future research in adapting similar techniques to other problems.

For this reason, I am inclined to accept the paper, but a higher recommendation should depend on its potential impact on the ICML audience.

**Questions For Authors:**

1. Please let me know if I missed any ML methodological contributions.
2. Are the authors planning to extend this work to other HN coloring problems or other discrete geometry problems?

**Relation To Broader Scientific Literature:**

I was really impressed by Davies et al. (2021), which demonstrated that AI can significantly advance mathematics. While its capability in formal proving is limited, at least for complex problems that cannot be easily derived using Lean, AI can assist humans in optimizing better solutions for combinatorics problems. This paper serves as another strong example of AI as a co-scientist, contributing to progress in discrete geometry problems.

**Theoretical Claims:**

Seems to be correct.

---

> ### Author Rebuttal · Authors · 2025-03-28
>
> Thank you for your positive assessment!
>
> We appreciate your thoughtful comments about our paper's position as a scientific contribution. You raise a fair point about methodological innovation versus application. We agree that it's always challenging to determine ML methodological contributions for these types of papers. In our view, progress often comes from either advancing methodology or applying reasonable/existing methods to new domains - we focused on the latter approach. This is similar to the work of [Davies et al. (2021b)](https://www.nature.com/articles/s41586-021-04086-x.pdf) and [Davies et al. (2021a)](https://msp.org/gt/2024/28-5/p06.xhtml), where the methodology of modeling statistical relations through neural networks and attributing relevance through gradients was not novel but nevertheless yielded interesting and relevant mathematical insights.
>
> > Are the authors planning to extend this work to other HN coloring problems or other discrete geometry problems?
>
> Yes, we definitely intend to continue applying our approach to other HN coloring problems and discrete geometry challenges, which we plan to include in any future journal version based on this submission.
>
> Thank you again for your review, please let us know if further clarification is needed.

---

### Official Review · Reviewer_uyCR · 2025-03-09

**Overall Recommendation:** 4

**Summary:**

The paper proposes an AI framework to tackle an open problem in coloring the plane in R^2. Some other variants consider R^n as well.
Authors reformulate the problem as an optimization task, they introduce a differentiable loss function and do greedy sampling to minimize the penalty loss they introduced.
Also authors discussed different variants of the same problem, describing exactly the loss to optimize again.
The work extends the d interval from where there exists a (1, 1, 1, 1, 1, d) coloring, a great advancement in the field.

**Claims And Evidence:**

Main claim is the authors improve the known distance range for colorings.

The authors state that constructions were "formally verified" in another venue (Anonymous, 2024), but given it's Anonymous we cannot verify this claim. Is there any other way to verify it?

**Essential References Not Discussed:**

Maybe some lines of work on AI4Math where AI (partially) solved open problems? There are a few to mention, but notably https://deepmind.google/discover/blog/alphageometry-an-olympiad-level-ai-system-for-geometry/ or https://proceedings.neurips.cc/paper_files/paper/2024/hash/aa280e73c4e23e765fde232571116d3b-Abstract-Conference.html

**Experimental Designs Or Analyses:**

Authors trained a simple MLP model, adding a sin activation and a final softmax. They claim they trained 100k runs, which, due to sampling and different training strategies is plausible.
They reported the AI result, which again seems very valid.
They could have added a bit more on lr, batch size to ensure reproducibility.

**Methods And Evaluation Criteria:**

This doesn't apply, given AI is used as a baseline for human to validate and prove formally some geometrical constructions.

**Other Comments Or Suggestions:**

N/A

**Other Strengths And Weaknesses:**

The framework’s adaptability to multiple variants is innovative. It would be interesting to see its applicability to other open research problems on geometric (or graph) coloring.

**Questions For Authors:**

The claim that this method "consistently recovered patterns resembling Pritikin’s construction" is not very quantifiable - we can visually inspect it, but there are some discrepancies. Is it possible to quantify them, or at least understand if the gap can be closed easily by human on harder problems?

What's the density of points/R^2 you tested?

**Relation To Broader Scientific Literature:**

The paper appropriately cites prior ML-for-math works (e.g., Karalias & Loukas, 2020; Fawzi et al., 2022) and connects to physics-informed NNs.

**Theoretical Claims:**

There aren't theoretical claims from the authors.

---

> ### Author Rebuttal · Authors · 2025-03-28
>
> Thank you for the valuable comments and taking the time to referee our submission! Let us address your points individually:
>
> > The authors state that constructions were "formally verified" in another venue (Anonymous, 2024), but given it's Anonymous we cannot verify this claim. Is there any other way to verify it?
>
> Regarding the anonymous citation, we would like to clarify that the referenced paper—submitted as supplementary material—was authored by us and provides a formal description of two colorings. It was published in a journal focused on combinatorial and geometric results. We consider formalized (and preferably published) constructions as the ultimate benchmark required to validate the methodology presented in this submission. However, conveying this clearly proved challenging due to the constraints of the double-blind review process. We are happy to allow our journal publication to be reviewed by the AC/SAC without disclosing our identities to the reviewers.
>
>
> > Maybe some lines of work on AI4Math where AI (partially) solved open problems? There are a few to mention, but notably https://deepmind.google/discover/blog/alphageometry-an-olympiad-level-ai-system-for-geometry/ or https://proceedings.neurips.cc/paper_files/paper/2024/hash/aa280e73c4e23e765fde232571116d3b-Abstract-Conference.html
>
> Thank you for pointing out the two papers. We are happy to broaden the scope of the related literature section and include them.
>
> >  It would be interesting to see its applicability to other open research problems on geometric (or graph) coloring.
>
> We fully agree. We are in the process of applying our methodology to other problems, but decided to focus on the Hadwiger-Nelson problem and its variants for this submission and reserved the application to other problems for future work.
>
> > The claim that this method "consistently recovered patterns resembling Pritikin’s construction" is not very quantifiable - we can visually inspect it, but there are some discrepancies. Is it possible to quantify them, or at least understand if the gap can be closed easily by human on harder problems?
>
> Regarding your question about recovering the results of Pritikin, this is unfortunately a point that is difficult to formally quantify. The neural networks provide numerical approximations that serve as visual inspiration for formal mathematical constructions. These numerical outputs typically achieve very low loss values in Equation (2), around 1e-7, whenever they correspond to actual formal constructions fulfilling the constraints of Equation (1). However, translating the suggested colorings into rigorous mathematical constructions is a manual and ad-hoc process requiring human interpretation and verification. This process varies in difficulty and for example required about a week’s worth of by-hand experimentation with pen and paper for Variant 2. In the case of Variant 1, we simply relied on visually inspecting the suggested colorings and the reported loss values and noting that they closely align with those described by Pritikin. Note that, after reviewing the data in Table 1, we found some inaccuracies in our values. These errors occurred because our training box was too small, causing the results to be affected by the cyclical pattern of the coloring. We corrected this by conducting the training in a much larger box. This correction produced values that are now much more consistent with Pritikin's / Croft's findings. We will make sure to update our submission to include the updated values. Regarding your questions of closing the gap when applying the approach to a harder problem: This depends heavily on the problem. In the case of Hadwiger-Nelson (in R^2), by its visual nature the problem can be approached by a human in a clear way. This may not be the case for other problems and could necessitate the application (or development) of other tools to construct formal solutions. For the Hadwiger-Nelson case (in higher dimensions), one could think of using Voronoi cells or other ways of creating polygons / parameterized geometrical shapes that closely match the output of the neural network but allow for a rigorous verification. In the general case, it is currently not possible for us to give a clear recipe on how to formalize the neural network outputs.
>
> > What's the density of points/R^2 you tested?
>
> Regarding your question about how many points we sampled for our numerical results, we evaluated the models on a grid of size 512 x 512 and uniformly sampled 256 points on a circle centered at each grid point. We will make sure to update our submission to include some more details regarding this in the appendix.
>
> Thanks again for your review, please let us know if further clarification is needed.

---

> > ### Comment · Reviewer_uyCR · 2025-04-05
> >
> > Thanks for the answer on the quantification - my only concern is really the one you mentioned, i.e. here in R^2 human can inspect the AI intuition and build on top. For R^4 this is not possible anymore, partially invalidating the help from AI. And this is why I wanted to know if we can measure the signal received by AI.

---

> > > ### Author Response · Authors · 2025-04-07
> > >
> > > Thank you for the follow-up. You're absolutely right that deriving formal constructions becomes significantly more challenging in higher dimensions. This is largely because our method doesn’t produce a clean discrete representation upfront—which could make formalization easier but at the cost of some flexibility. Instead, we use a more continuous approach. Still, there is signal: the numerical outputs often suggest that something nontrivial is happening, and in some cases, even simple discretizations can yield valid formal bounds. While interpreting results in R^3 or R^4 is undeniably harder, it’s not impossible—just more demanding. Even in R^2, we found it helpful to build small ad-hoc tools and perform targeted analysis to extract formal constructions. For higher dimensions, we expect similar, albeit more advanced, tooling and mathematical creativity to play a key role (which we are currently in the process of building). We’re happy to include a brief discussion of this in the appendix when describing how the two formal constructions were derived. From an ML perspective, the fact that it is not an end-to-end procedure may seem somewhat unsatisfactory, but we see real value in this collaborative interaction which enables deeper mathematical exploration—AI offers new intuition, and that can be developed further using traditional mathematical techniques.

---

### Official Review · Reviewer_cNSg · 2025-03-12

**Overall Recommendation:** 5

**Summary:**

The paper proposes a neural-net approach to solving the Hadwiger-Nelson problem. The problem involves coloring the plane under the constraint that no pair of points with unit distance has the same color. The paper treats this problem as an unsupervised combinatorial/continuous optimization problem. By adopting a probabilistic formulation of the problem, the authors design a differentiable loss function and use gradient descent to train a neural net to minimize the constraint violation for randomly sampled points in space.

The approach is flexible and enables solving several variations of the problem including a version that allows for different distance constraints on each color as well as coloring in higher dimensions. and is able to find new valid colorings for different variants of the problem.

**Claims And Evidence:**

Some details are unclear. Some of the limitations of the approach are not adequately discussed.
- The paper does not clearly explain how the 'formalized' colorings are obtained from the informal ones. There seem to be some subtleties around this that should be carefully discussed. For example, the numbers in table 1 suggest that the neural coloring proposed is an improvement. On the other hand, the comments in the table caption say that there are similarities with known constructions that prevent this coloring from being formalized. I believe observations like this merit further discussion and a more thorough explanation of the process of converting a 'neural' coloring to a formalized one should be given (perhaps in the appendix).

**Essential References Not Discussed:**

There's this paper on constructing aperiodic tilings with GNNs in an unsupervised fashion [1] which is somewhat similar in spirit to this one. The loss function isn't as rigorously constructed but I believe it should be mentioned in your paper.


1. Hao Xu, Ka-Hei Hui, Chi-Wing Fu, and Hao Zhang. 2020. TilinGNN: learning to tile with self-supervised graph neural network. ACM Trans. Graph. 39, 4, Article 129 (August 2020), 16 pages. https://doi.org/10.1145/3386569.3392380

**Experimental Designs Or Analyses:**

See my other comments.

**Methods And Evaluation Criteria:**

The benchmarking setup is clear. However, I recommend constructing a table that summarizes the results of the paper for each variant of the problem. One column could include the best-known result for the given variant (and a citation to the corresponding paper) and another with the best achieved by the proposed method. The table (or its caption) could also refer to specific figures or sections in the paper and/or provide additional context.

**Other Comments Or Suggestions:**

N/A

**Other Strengths And Weaknesses:**

Overall, I like the paper and I think it provides a rather interesting perspective on theorem proving with the help of neural nets. The approach is fairly simple and it shows that one can leverage neural nets to naturally parameterize problems in math. I begin with a tentative score that I am willing to increase once some of my concerns and questions have been addressed.

**Questions For Authors:**

N/A

**Relation To Broader Scientific Literature:**

The findings relate to mathematical discovery and ML-assisted theorem proving. The paper establishes new colorings that were previously not discovered.

**Theoretical Claims:**

- The authors use the predictions of the neural network as plausible candidate patterns for formalized colorings. Indeed, the paper claims that strictly speaking, one cannot guarantee that the colorings discovered in a given box generalize to the entire plane.  What are the main obstructions when it comes to generalizing a pattern from a fixed box to the entire plane? Are there some known conditions?

- As a follow-up to the question above, are there any conditions on the size of the box? I am asking this because it seems that a really small box (say [-1e-6, 1e-6] might introduce trivial colorings. Or is that not the case and why?

- Proposition 1 suggests that a valid probabilistic mapping will be found for points in the box and their neighbors in the plane. The integral for the loss is estimated by appropriately sampling points and minimizing the constraint violations for them. My question is: I suspect the loss is never quite 0, even for just the subset of points that the neural network sees in training. To get the results you demonstrate in the images, what values of the loss are sufficient? Are they really small/close to 0? Does the neural net converge to some small number that is clearly above zero?  Moreover, what happens when you use the Lagrangian relaxation with a specific coefficient? Did you have to eyeball the solutions in this case?

---

> ### Author Rebuttal · Authors · 2025-03-28
>
> Thank you for your positive and very relevant comments! Let us split our response into three parts.
>
> **The relationship between the NN output and formal results**
>
> You point out an important aspect that we could have perhaps made more clear in our submission. The neural networks provide numerical approximations that serve as visual inspiration for formal mathematical constructions. These numerical outputs typically achieve very low loss values in Equation (2), around 1e-7 evaluated on a discretized grid of size 512 x 512, whenever they correspond to actual formal constructions fulfilling the constraints of Equation (1). However, translating the suggested colorings into rigorous mathematical constructions is a manual and ad-hoc process requiring human interpretation and verification. This process varies in difficulty:
>
> 1. For the triangles (Variant 4), formalization was relatively straightforward as the networks simply suggested stripe-based colorings with previously not considered widths.
> 2. For off-diagonal colorings (Variant 2), formalization was more complex though ultimately only relied on basic trigonometry and about a week’s worth of by-hand experimentation with pen and paper.
> 3. For 3D colorings (Variant 3), formalization remains challenging despite promising numerical results. We are exploring approaches using Voronoi cells. The paper by Xu et al. that you pointed out may also prove relevant in this context, so thank you for that recommendation!
>
> We should point out that this manual process is also why we chose to publish the actual formal constructions as separate results in journals more suited to these types of questions while choosing ICML as the venue where we describe the underlying machine-learning-guided framework that led to these discoveries, similar to the work of [Davies et al. (2021b)](https://www.nature.com/articles/s41586-021-04086-x.pdf) on knot invariants that was separately formalized by [Davies et al. (2021a)](https://msp.org/gt/2024/28-5/p06.xhtml). We will make sure to more clearly highlight this aspect in the revision of the paper.
>
>
> **Main achievements and tabular summary**
>
> Firstly, we see the main achievements obtained using the methodology described in this paper as follows:
>
> 1. **Variant 1: “Re-discovered” previously known constructions due to Pritikin/Part (Table 1, Figures 4, 10).**  Note that, after reviewing the data in Table 1, we found some inaccuracies in our values. These errors occurred because our training box was too small, causing the results to be affected by the cyclical pattern of the coloring. We corrected this by conducting the training in a much larger box. This correction produced values that are now much more consistent with Pritikin's / Croft's findings.
> 2. **Variant 2: Clear and formalized improvements in the form of the two colorings (Figure 2, 12).** The description of these was published independently in a journal specializing in combinatorial and geometric problems.
> 3. **Variant 3: Suggests 3D space can "almost" be colored with 14 colors.** As already mentioned, the description of a formal result is still work-in-progress.
> 4. **Variant 4: Clear and formalized improvements (Figure 3).** We intend to submit the description of these to a journal specializing in combinatorial and geometric problems.
>
> We should emphasize that we view formalized (and preferably published) constructions as the ultimate benchmark required to validate the methodology presented in this submission. This, as well as the fact that improvement is often not measurably through a single metric, makes summarizing the results in a tabular environment a bit challenging. Your point is still valid and well received though and we will make sure to summarize and emphasize the results more clearly in our submission.
>
>
> **Additional points**
>
> * We tested values between 1e-5 and 5e-1 for the Lagrangian coefficient. Values from 1e-3 upwards led to runs where the constraint was (effectively) fulfilled. There was however not a clear "sweet spot" or tendency for larger weights to produce better colorings. We simply treated this as a hyperparameter that needed to be tuned.
> * There is no guarantee that patterns from a finite box extend to the entire plane, though all our promising candidates exhibited clear periodic structures. Boxes with side lengths anywhere between 6 and 10 units proved sufficient for clear patterns to emerge without computational overhead. Choosing a box that is too small would certainly lead to non-sensical results that do not relate to any coloring of the plane.
> * The paper by Xu et al. that you suggested seems very interesting and highly relevant and we will make sure to include a reference to it.
>
> We would like to thank you again for your positive and constructive review. We hope to have addressed your concerns, please let us know if further clarification is needed.

---

> > ### Comment · Reviewer_cNSg · 2025-04-07
> >
> > OK, thank you for your response. Please make sure to discuss the sensitivity to parameters like the size of the box. I mentioned it in my review originally and it seems that the first item in your list suggests that it indeed plays an important role. Overall, I like the paper and my only issue is that it does seem that some heavy lifting needs to be done on the formalization after the network provides a candidate coloring. In any case, I think this is a creative use of neural networks that will definitely be interesting to the broader ML community. I increased my score.

---

> > > ### Author Response · Authors · 2025-04-09
> > >
> > > Thank you again for your valuable review and for increasing your score. We will certainly discuss the sensitivity to the hyperparameters and the required formalization work more clearly.

---

### Decision · Program_Chairs · 2025-05-01

**Decision:**

Accept (oral)

**Comment:**

This is an excellent paper contributing to the emerging but still challenging field of using AI methods to derive new mathematical research. While the mathematical findings (new constructions for the Hadwiger-Nelson problem) have been peer-reviewed and published in a mathematical journal, this paper focuses on the underlying ML methods that were used to obtain mathematical intuitions that were the base of the novel constructions.
How to successfully use AI for math research remains a big challenge due to the unique rigorous and complex nature of mathematics. While there have been a handful previous works demonstrating that pairing up AI (to deliver intuitions) and math researchs (to make them rigorous) is a useful strategy, we are in desperate need of more success stories of that kind. This paper is such a success story.